# Shelf-Life Evaluation of “San Marzano” Dried Tomato Slices Preserved in Extra Virgin Olive Oil

**DOI:** 10.3390/foods10081706

**Published:** 2021-07-23

**Authors:** Vincenzo Sicari, Mariarosaria Leporini, Rosa Romeo, Marco Poiana, Rosa Tundis, Monica Rosa Loizzo

**Affiliations:** 1Department of Agraria, “Mediterranea” University of Reggio Calabria, Cittadella Universitaria, 89124 Località Feo di Vito, RC, Italy; vincenzo.sicari@unirc.it (V.S.); rosa.romeo@unirc.it (R.R.); mpoiana@unirc.it (M.P.); 2Department of Pharmacy, Health and Nutritional Sciences, University of Calabria, 87036 Rende, CS, Italy; mariarosarialeporini@tiscali.it (M.L.); rosa.tundis@unical.it (R.T.)

**Keywords:** tomato slices preserved in extra virgin olive oil, food analysis, phenols, carotenoids, CIELab parameters, antioxidant activity

## Abstract

Since ancient times, vegetables have been preserved in oil, to be consumed throughout the year, and not just during the period in which they were harvested. Dried tomato slices in Extra Virgin Olive Oil (EVOO) are one of the most famous Italian preserves. This is the first study which aimed to investigate the shelf-life parameters of this preserve during the 12 months of storage in both light and dark conditions. For this purpose, quality and CIELab color parameters were analysed in EVOO alone and as preserving liquid; total phenols and carotenoids content as well as β-carotene and lycopene content, a fatty acids profile, and antioxidant activities were examined. Results showed that samples stored in the dark are protected against degradative processes. Moreover, after 6 months of storage, the EVOO used as preserving liquid is enriched by the phytochemicals contained in dried tomato slices. This enrichment of EVOO by tomato bioactive compounds is reflected in the increase in the antioxidant activity of the oil independently by the presence of light during storage.

## 1. Introduction

For centuries, people have sought ways to preserve their food and thus reap the benefits of a seasonal harvest throughout the year. In many Mediterranean countries, one of the most natural ways to preserve foods, such as vegetables, meats, fish, cheese, and herbs, involved and still involves the use of olive oil [1]. Olive oil prevents spoilage by isolating food from air, providing a seal that can delay oxidation, deterioration, and formation of mould. Preserving in olive oil is particularly well suited to certain foods that are eaten in small quantities, such as sun-dried tomatoes, sweet peppers, or mushrooms. Virgin olive oil is the main fat of the Mediterranean diet. Furthermore, it is considered to be a functional food due to its content of bioactive phenols [2].

The dietary consumption of extra virgin olive oil (EVOO) has a number of benefits: antioxidant properties and anti-inflammatory, anti-cancer, and cardio-protective effects [3,4].

Furthermore, hydroxytyrosol, pinoresinol, and oleuropein demonstrated to possess anti-microbial activity. Pinoresinol has shown antifungal activity against several pathogenic fungi such as *Fusarium verticillioides*, *Fusarium graminearum*, and *Candida albicans* [3].

The quality of EVOO is related to several factors, such as olive variety, time and temperature of storage, the presence of oxygen, and the packaging material [5,6].

Tomato (*Lycopersicon esculentum*) is cultivated worldwide. It is the second most important vegetable crop after the potato [7]. The “San Marzano” tomato is a protected designation of origin (PDO) variety. This typical Italian product derived from Agro-Sarnese-Nocerino area (Campania, Italy). “San Marzano” was characterized by an intense red color and unique consistency that makes it resistant to processing [8]. This variety has been poorly investigated. It is well known that the tomato has been found to possess many health benefits, particularly to prevent prostate, cervical, stomach, rectal, pharyngeal, and oesophageal cancers [9]. Post-harvest decay of vegetables is currently a major challenge for researchers working in the food sector. Often, tomatoes produced in the peak seasons are either consumed fresh, sold at relatively cheap prices, or allowed to go waste [10]. To avoid waste and to enjoy the tomatoes all year round, farmers in southern Italy dry them in the sun before storing them in glass jars with EVOO as the preserving liquid. These sun-dried tomatoes can be eaten on “bruschetta” or used to flavor pasta dishes and salads. They are part of Italy’s gastronomic heritage. Sun-dried tomatoes are very easy to prepare, especially in the hot, dry, summer climate of southern Italy. In other climates, the tomatoes, picked at peak ripeness, may be dried using an oven or dehydrator. Most store-bought dried tomatoes have been prepared in this way.

The concept of food quality is quite complex. In fact, in recent years, quality that is intended as chemical and hygienic has become the set of different factors such as size, shape, color, texture, flavor and also an acceptable content in bioactive compounds. For this reason, in order to monitor the shelf-life of the product, the most appropriate parameters must be selected from time to time.

The European Union legislation EEC Reg. 2568/91 and subsequent amendments (EC Reg. 61/2011) has set the minimum quality standards that the oil must present in order to be marketed as EVOO [11,12]. In particular, the parameters considered to evaluate the EVOO quality are acidity, peroxide number, spectrophotometric characteristics, and phenolic compounds content, which influence the organoleptic profile, fatty acids, and pigments which are responsible for the color of the oil and are partly involved in the oxidative mechanisms [6]. Appearance of food strongly influenced a consumer’s opinion about the food quality. Food color is linked to the chemical, biochemical, microbial, and physical changes which occur during maturation, postharvest handling, and processing. The evaluation of food color by human perception could vary by persons and the environment-like lighting condition at the place, etc. Industrial food product quality was monitored and controlled by an imaging system based on CIELab parameters [13].

This study aimed to investigate, for the first time, the shelf-life parameters of dried tomato slices preserved in EVOO during 12 months of storage, in both light and dark conditions. This aspect is very important due to the sensitivity of EVOO to alteration, and the subsequent transmission to dry tomatoes of negative characters derived from oil rancidity. For this purpose, EVOO oil from preserves in both storage conditions and dried tomato slices were subjected for 12 months to a wide quality characterization of the most valuable indexes.

## 2. Materials and Methods

### 2.1. Chemicals and Reagents

All reagents were purchased from Sigma-Aldrich S.p.a. (Milano, Italy), whereas analytical-grade solvents were obtained from VWR International s.r.l. (Milan, Italy). The microbiological reagent Plate Count Agar medium was purchased from Conda-Pronadisa, (Spain) and the Dichloran Rose-Bengal Chloramphenicol agar medium was obtained from VWR Chemicals s.r.l. (Milan, Italy).

### 2.2. Sample Preparation and Experimental Design

This study was carried out on the “Ottobratica” cultivar grown in the experimental orchards belonging to the Calabrian regional government, located in Gioia Tauro, Reggio Calabria (Italy).

The olives were harvested by hand in mid-October 2019 directly from the plant, and immediately taken to the laboratory, where the oil was extracted within 12 h of harvesting, using a crusher “Mini 30” made by Agrimec (Firenze, Italy).

After crushing and malaxing of the olive paste, oil extraction was performed at room temperature by means of a pressure system. The olive paste was mixed for 30 min, and then a pressure of 200 atm was applied for 40 min. The oil was separated by centrifuge from the water, then filtered through a paper at room temperature and immediately analysed. EVOO was stored at 10 °C in the dark using green glass bottles without headspace before analysis. EVOO was analysed one month after production and employed to preserve dried tomatoes.

Fresh “San Marzano” tomato fruits produced in a farm located in Nocera, Salerno (Italy) devoid of any form of known injury were obtained from market in Reggio Calabria (Italy), and were transported in a polythene bag to the laboratory at the Department of Agraria of Mediterranean University. Two kg of “San Marzano” tomato were washed with chlorinated water and cut in thin slices with a sterilized stainless-steel knife to obtain a slice about 3–4 mm thick. After, the samples were distributed uniformly as a thin layer on the stainless-steel trays of size 0.3 × 0.2 m, sprinkled with salt, and dried in an oven at 35 °C. The dried tomatoes were placed in a glass jar, and pressed to be stacked. During the pressing, the EVOO was added to cover all the tomatoes. The containers were then hermetically sealed. The jars were submitted to pasteurization at 80 °C for 15 min and then cooled at room temperature [14]. In particular, these parameters were employed: (i) 40 jars contained EVOO alone; (ii) 40 jars contained EVOO used as the preserving liquid of dried tomatoes (300 g) in light condition storage; (iii) 40 jars contained EVOO used as the preserving liquid of dried tomatoes (300 g) in dark condition storage. The monitoring was carried out on a monthly basis for 12 months. The acronyms of the analysed samples are shown in Table 1.

### 2.3. Quality and CIELAB Parameters

The free acidity value, peroxide index, UV light absorption (K232 and K270), and ΔK of samples were analysed according to the methods described by EC Regulation [15]. Water activity (aw) of the tomato was measured by Aqualab LITE hygrometer (Decagon devices Inc., Washington, DC, USA).

The moisture content of tomato slices samples was determined by 934.06 AOAC [16]. Five grams of tomatoes were weighed into the moisture dishes. Samples were oven dried at 105 °C for 3 h to a constant weight and then transferred quickly into a desiccator for cooling. After cooling, the samples were weighed while minimizing exposure to atmospheric conditions. The moisture determination was done in three replicates for each sample. Moisture content was obtained through differences in weight before and after moisture drying to a constant weight. The values obtained were expressed as percentage moisture content.

Chromatic coordinates were measured at 25 °C using a PCE CSM-4 colorimeter (PCE, Lucca, Italy) to obtain the color according to the CIEL a* b*method [17]. Data were expressed as higher saturation of color or Chroma (C*). Hue angle (h*), considered the qualitative indicator of color, is an attribute according to which colors have been traditionally defined as reddish, greenish, and is used to define the difference of a certain color with the reference to a grey color of the same lightness. This attribute is related to the differences in absorbance at different wavelengths. A higher hue angle represents a lesser yellow character in the assays. An angle of 0° or 360° represents red hue, whilst angles of 90°, 180°, and 270° represent yellow, green, and blue hues, respectively [18].
Hue angle (h*) = tan^−1^ (b*/a*)(1)
(2)$$\mathrm{Chroma}\;(C*)=\sqrt{a^{*}{}^{2}-b^{*}{}^{2}}$$

In order to evaluate the overall change of color, the ΔE*ab parameter was calculated following the equation [19]:(3)$$\Delta E*\mathrm{ab}=\sqrt{{\left(L_{0}^{*}-L^{*}\right)}^{2}+{\left(a_{0}^{*}-a^{*}\right)}^{2}{\left(b_{0}^{*}-b^{*}\right)}^{2}}$$
where the index “0” indicates the sample at time 0, whereas the letters without the index correspond to the parameters of the samples during storage.

### 2.4. Microbiological Analysis

The microbial population of samples was analysed following the standard procedures of serial dilution and plate count as described by Akpan et al. [20] with some modifications. Five g of each sample were weighed and placed in a stomacher bag, and it was diluted in 90 mL of sterilized water and homogenized using a stomacher (Model 400 circulator, Seward, Norfolk, UK). Samples were analysed for aerobic mesophilic bacteria with surface inoculation (1 mL/plate, duplicate) in PCA (Plate Count Agar). DRBC (Dichloran Rose Bengal Chloramphenicol) was used for the enumeration of yeasts and moulds, and the plates were inoculated at 26 °C for 4–5 days before counting the colonies.

Each test was done in duplicate.

### 2.5. Fatty Acid Profiles

The EVOO, EVOO_DTD, and EVOO_DTL fatty acids profile was investigated by gas chromatography (GC) and gas chromatography-mass spectrometry (GC-MS), after their derivatization to methyl esters, as previously reported [21]. Analyses were carried out on a Shimadzu GC17A (Shimadzu, Milan, Italy) equipped with a flame ionization detector (FID) and an HP-5 MS (30 m × 0.25 mm × 0.25 µm) column. Nitrogen was used as the carrier gas (flow rate of 1.0 mL/min). Data were acquired by using the Borwin Software.

The oven temperature programming was 60 °C during injection, and then increased from 60 to 280 °C at the rate of 14 °C/min. FAMEs were identified using a Hewlett-Packard 6890 gas chromatograph (Agilent, Milan, Italy) equipped with an HP-5 MS capillary column and a Hewlett Packard 5973 Mass Selective (EI, 70 eV) (Agilent, Milan, Italy). The same conditions described for the GC analyses were used. Identification of peaks corresponding to FAMEs was accomplished by means of a standard mixture of FAMEs purchased from Supelco and by reference to the Wiley 138 mass spectra library.

### 2.6. EVOO and Dried Tomato Extraction Procedure

The EVOO phenolic extract was obtained following the procedure proposed by Montedoro et al. [22] with some modification. In brief, EVOOs (5 g) were extracted with 2 mL of methanol (MeOH)/water (7:3, *v*/*v*), and then 1 mL of *n*-hexane was added. The mixture was centrifuged at 12,000 rpm for 10 min at 4 °C. The extracts were pooled and evaporated at low temperature. The dry extracts were re-suspended in 1 mL of methanol.

The dried tomatoes were subject of extraction with the ultrasound-assisted maceration procedure using ethanol (EtOH). For this extraction procedure, three extraction cycles (3 × 1 h) with an ultrasonic frequency of 40 kHz at a temperature of 30 °C for 60 min were conducted for each sample in a Branson model 3800-CPXH water bath (Branson, Milan, Italy). After each extraction cycle, the mixture was filtered through Whatman filter Paper 4 under vacuum, and the solvent was removed using a rotary vacuum evaporator at 30 °C. Each extraction was performed in triplicate.

For chlorophyll determination, dry tomato slices (1 g) were grinded and subjected to maceration with N with 80% acetone [23].

### 2.7. Total Phenol, Flavonoid, Carotenoid and Chlorophyll Contents

Total phenol content (TPC) was evaluated by using the Folin-Ciocalteu method, as previously reported [24]. Samples at the concentration of 1.5 mg/mL (0.1 mL) were mixed with a solution of Folin-Ciocalteu reagent (0.5 mL) and water (1 mL). After 1 min of incubation, 1.5 mL of 20% sodium carbonate was added, and the mixture was incubated at room temperature. The absorbance was measured at 765 nm using a UV-Vis (Jeneway 6003, Carlo Erba, Milan, Italy). TPC was expressed as mg of chlorogenic acid equivalents (CAE)/g of the sample.

The total flavonoid content (TFC) was determined using a method based on the formation of a flavonoid-aluminium complex [25]. Samples were mixed with aluminium chloride solution (2%) in a 1:1 ratio and incubated at room temperature for 15 min. The absorbance was measured at 510 nm, and TFC was expressed as mg quercetin equivalents (QE)/g.

The total carotenoid content (TCC) was determined, as previously described [26]. Briefly, 1 mL of the extract was added to 0.5 mL of NaCl 5% solution, vortexed for 30 s, and centrifuged at 4500 rpm for 10 min. The supernatant (100 µL) was diluted with 0.9 mL of *n*-hexane and measured at 460 nm. TCC was expressed as mg/Kg.

The total chlorophyll content was estimated by measuring the absorption at 662 and 644 nm, and calculated according to the procedure previously published by Ašimović et al. with some modifications [23]:Chlorophyll a = [12.7 (A_662_) − 2.69 (A_644_)] × V/100 × w(4)
Chlorophyll b = [22.9 (A_644_) − 4.86 (A_662_)] × V/100 × w(5)
Total chlorophyll = [20.2 (A_644_) + 8.02 (A_662_)] × V/100 × w(6)
where: V, Total volume of solution made (mL); w, weight of sample (g); A_644_, Absorbance at 644 nm; A_662_, Absorbance at 662 nm. The chlorophyll content was expressed as mg/Kg.

### 2.8. Evaluation of Lycopene and β-Carotene Content

The spectrophotometric carotenoid determination was carried out after extraction. Before extraction, samples were homogenized in a blender, and an aliquot of 10 g of the sample was weighed into a 200 mL amber colored flask wrapped with aluminum foil. The analyses were carried out in darkness to prevent carotenoid degradation and isomerization. A total of 100 mL of the solvent mix (*n*-hexane/acetone/methanol 2:1:1 *v*/*v*/*v*) was added to the flask and sonicated continuously for 10 min (Misonix Ultrasonic Liquid Processor, USA). The extraction was repeated until the sample became colorless. The combined extract was transferred to a separating funnel, and 5 mL of distilled water were added to separate polar and nonpolar phases. The nonpolar hexane layer containing carotenoids was collected and concentrated in a rotary evaporator (Heidolph, Germany) until dryness. The residue was dissolved in *n*-hexane (10 mL). Lycopene and β-carotene contents were determined according to Fish et al. [27] using a UV-Vis spectrophotometer (Agilent 8453 Technologies, Italy). To minimize the interference from other carotenoids, the concentration of lycopene was calculated at λ = 503 nm using the molar extinction coefficient ε = 17.2 104 M^−1^·cm^−1^.

For β-carotene, the absorbance was measured at λ = 450 nm, and the quantification was carried out using a standard curve. Results were expressed as mean ± standard deviation (SD) (*n* = 3).

### 2.9. Antioxidant Activity

The in vitro antioxidant activities of samples were evaluated by using ABTS, DPPH, and β-carotene bleaching assays. The ABTS radical scavenging test was done following the procedure reported by Leporini et al. [28]. The ABTS solution (7 mM) was mixed with potassium persulphate (2.45 mM) to obtain a solution of ABTS^+^ radical cation. After 12 h, the latter solution was diluted with ethanol to a final absorbance of 0.70 at 734 nm employing a UV–Vis Jenway 6003 spectrophotometer. Successively, 2 mL of diluted ABTS+ solution were added to extracts (25 μL) at concentrations from 400 to 1 μg/mL. After 6 min, the absorbance was read at 734 nm. The ABTS scavenging capacity was calculated as follows: ABTS scavenging activity (%) = [(A_0_ − A)/A_0_] × 100, where A_0_ is the absorbance of the control reaction, and A is the absorbance in the presence of extract.

The DPPH radical scavenging test was applied using the procedure previously published [29]. Samples at different concentrations (1–1000 μg/mL) were added to DPPH solution (1.0 × 10^−4^ M), and after 30 min the absorbance was measured at 517 nm. The DPPH radical-scavenging activity was calculated following the equation: DPPH radical-scavenging activity (%) = [1 − (sample absorbance with DPPH − sample absorbance without DPPH)] × 100. Ascorbic acid was used as positive control in both radical scavenging assays.

The potential of samples to inhibit lipid peroxidation was assessed using the β-carotene bleaching test [29]. In brief, Tween 20, linoleic acid, and β-carotene were mixed. After evaporation of the solvent, the emulsion was added into the 96-well microplate containing samples at different concentrations (2.5–100 μg/mL). After incubation at 45 °C for 30 min, the absorbance was measured at 470 nm. Propyl gallate was used as a positive control.

### 2.10. Statistical Analysis

Data are expressed as means of three different experiments ± standard deviation (S.D.). GraphPad Prism version 4.0 for Windows (GraphPad Software, San Diego, CA, USA) was used to calculate the concentration giving 50% inhibition (IC_50_) by plotting the percentage inhibition versus concentration. Pearson’s correlation coefficient (*r*) was calculated by using Microsoft Excel 2010 software. Differences within and between groups were evaluated by the one-way analysis of variance test (ANOVA) followed by a multicomparison Dunnett’s test (α = 0.05) that was used to compare each group with the positive control in biological assays at different levels and Tukey’s test to determine any significant difference in chemical parameters among investigated samples. Principal Component Analysis (PCA) was applied by SPSS software for Windows, version 15.0 (Chicago, IL, USA). Statistical analyses were performed using SPSS software for Windows 22.0 Version (SPSS Inc., Elgin, IL, USA). Relative Antioxidant Capacity Index (RACI) was applied to evaluate the antioxidant capacity values generated by different in vitro methods [30].

## 3. Results and Discussion

### 3.1. Evolution of Dried Tomatoes Preserved in Oil Quality Parameters during Shelf-Life

The acidity of EVOOs was used as a reference parameter for the overall evaluation of the product quality. The highest quality EVOOs must feature a free acidity lower than 0.8%. In the first month, the samples maintain constant acidity values (0.68%), while after the fourth month, a significant increase was observed. In particular, at 12 months, values of 0.98%, 0.97%, and 0.94% were found, for EVOO, and EVOO_DTL, and EVOO_DTD, respectively (Table 2). It is clear that the presence of the tomatoes slices into EVOO and storage in the dark helped protect oil from degradation.

Oxidation causes the product to become rancid, forming unpleasant odors and flavors. According to current regulations, the maximum number of peroxides is fixed at 20 milliequivalents of oxygen per kg of oil (meq O_2_/kg). Reduced levels of peroxides are associated with an increase in the shelf-life of the product and a delay of rancidity. Peroxide formation is related to the action of oxygen, high temperatures, light, and the catalytic action of some enzymes capable of binding oxygen to the fatty acids.

A large number of peroxides indicate the start of an irreversible oxidative process. After 12 months, an increase in the levels of peroxides was observed, with values of 22.54 and 26.24 meq O_2_/kg in EVOO and EVOO_DTL, respectively (Table 2). Less variation from 15.55 to 20.71 meqO_2_/kg was observed for EVOO_DTD, suggesting that the presence of tomatoes and storage in the dark reduced degradation and improved the shelf life of the products. All EVOO samples exhibited initial ΔK values of 0.01. This parameter was maintained (0.17 and 0.16, respectively) with a minimal variation in EVOO_DTL and EVOO_DTD.

Concerning dried tomatoes, water activity (aw) and moisture content were also considered (Appendix A). Water activity expresses the relationship between the vapor pressure of the water in the food and the vapor pressure of pure water. It is a dimensionless value that indicates the amount of water contained in the product, free from bonds with other components, and capable of giving chemical and biological reactions, and available for food enzyme activity and microbial growth [31].

When the aw value increases, the shelf life of the product decreases. In general, foods present aw levels in the range of 0.2 (very dry foods) to 0.99 (moist fresh foods). Micro-organisms maintained their viability regardless of the aw, but to grow bacteria requires an aw > 0.8, whereas yeasts and moulds require an aw > 0.6 [31]. Obtained data relating to moisture content and aw showed that when stored in the light, moisture content varies from 31.66 to 40.4%, and aw from 0.63 to 0.71. However, when the product was stored in the dark, a less significant increase was observed. After 12 months, a moisture content of 37.63% and an aw of 0.69 were found (Appendix A). These results demonstrated that the absence of light is a fundamental parameter to extend the shelf life of the product.

Recently, Owureku-Asare et al. [31] compared the physicochemical characteristics of cabinet dryer (called solar cabinet dryer) and sun-dried tomato powder. Results showed that the moisture content of the solar dried tomato (13.94–14.57%) was lower than the sundried tomato (19.38–21.63%). Additionally, the same authors reported aw values lower for solar cabinet dried tomato powder (0.35–0.38) compared to sundried tomato powder (0.53–0.57).

During storage, EVOO loses its quality (expressed as free acidity and peroxide content) with respect to samples containing dry tomatoes (EVOO_DTL and EVOO_DTD). Particularly EVOO_DTD showed a lesser increase in these two parameters.

### 3.2. Evolution of Phytochemicals in Dried Tomatoes Preserved in Oil

Carotenoids and chlorophylls, extracted from the drupe during the transformation process, are natural pigments that influence the EVOO color. For this reason, we decided to monitor the chlorophyll content, total carotenoids content (TCC), and β-carotene and lycopene contents of investigated samples. The CIELab parameters were also investigated.

The most significant chlorophyll content change was observed in EVOO_DTL with variation from 66.17 to 23.10 mg/Kg, followed by EVOO with a chlorophyll content ranging from 66.17 to 28.65 mg/Kg (Appendix A). A similar situation was observed in EVOO_DTD where a reduction of the content in this pigment was verified, and the storage conditions in the dark better preserved the matrix from degradation. In fact, chlorophyll content decreased by −38%, −43%, and −37%, respectively for EVOO, EVOO_DTL, and EVOO_DTD.

A significant reduction in TCC was observed in EVOO with values in the range from 22.45 to 11.78 mg/Kg, while the TCC level remained quite constant in EVOO_DTL (22.45–21.35 mg/Kg) (Appendix A). After 12 months, a reduction in the carotenoid content of 10.7 and 1.1% was observed for EVOO and EVOO_DTL, respectively. On the contrary, there is a notable increase (+2.5%) in EVOO_DTD due to the contribution provided by the dried preserved tomato. Indeed, the carotenoid content in DTD decreased from values of 24.57 to 19.96 mg/Kg. A significant reduction of −6% in the TCC was observed in DTL.

The lycopene content of EVOO_DTL and EVOO_DTD found at month 0 was probably caused by the applying of manual pressure to stack the dried tomatoes inside the jar containing the oil used for preservation. This pressure may facilitate the extraction of lycopene by the EVOO (solvent) as a consequence of the maceration process, taking into account that analysis was carried out 72 h after jar preparation [32]. Analysis of oils (EVOO_DTL and EVOO_DTD) evidenced that in the first 6 months of storage, a decrease in lycopene content was observed. Successively, this content increased due to the passage of this pigment from the dried pressed tomatoes to the EVOO. After 12 months, an increase in the lycopene content of 0.70% and 0.20% was observed, for EVOO_DTD (0.91–1.59 mg/100 g) and EVOO_DTD (0.91–1.10 mg/100 g), respectively. On the contrary, the lycopene content in DTL and DTD decreased by 0.31% and 0.49%, respectively, with a content variation from 0.98 to 0.67 mg/100 g and from 0.98 to 0.4 mg/100 g, respectively. Moreover, we have to consider that tomato’s lycopene content could be influenced by several factors, such as the tomato genotype, plant nutrition, the environment, and the period of collection [33]. Previously, Karakaya and Yılmaz [34] investigated the lycopene content in fresh and processed tomatoes and found values of 1.74, 5.51, and 3.55 mg/100 g in fresh, sun-dried, and canned tomatoes, respectively. The lycopene content of fresh “Roma” tomatoes found by Owureku-Asare et al. [31] was 1.49 mg/100 g. De Abreu et al. [35] reported a lycopene content in the range of 19.97–33.95 mg/100 g dried material (DM) in eight preserved (in oil) dried tomatoes. These results are agreement with Toor and Savage [36]. Similar lycopene values were found in cherry tomatoes dehydrated at different temperatures [37]. Values in the range of 23.05 to 29.77 mg/100 g DM were observed with the highest value in tomatoes subjected to dehydration at a low temperature (40 °C). Additionally, Mwende et al. [38] suggested that lycopene content is correlated to the tomato variety. These authors studied four varieties of fresh tomatoes, grown under the same conditions, and found lycopene contents of 174.86, 108.46, 135.80, and 198.25 mg/100 g DW for “Anna F1”, “Kilele”, “Prostar F1”, and “Riogrande”, respectively. Different factors, such as oxygen, high temperatures, and light, reduced the TCC in this food matrix [39].

A great variability in the β-carotene content was found (Table 3). In EVOO, the content decreased from 90.26 to 38.61 mg/100 g of oil, while in EVOO_DTL and EVOO_DTD, less significant reduction (−17% and −4%) occurred after 12 months due to the protective action exerted by tomatoes. Indeed, a reduction of 5% in β-carotene content was found in both DTD and DTL (Table 4). De Abreu et al. [35] reported a β-carotene content in the range of 210–447 mg/100 g DM found in eight preserved (in oil) dried tomatoes.

The variation in pigment content also influenced CIELab parameters (Table 5). After 12 months of storage, the L* parameter was highest in the EVOO samples followed by EVOO_DTL which could be due to auto-oxidation and subsequently to a greater degradation of chlorophylls in these samples. On the contrary, the C* value was highest in EVOO_DTD and EVOO_DTL due to a red color conferred by the lycopene contained in dried tomatoes (Table 5). No significant differences were recorded in Hue (H*) values.

Interesting Pearson’s correlations were found. In EVOO, C* positively correlated with chlorophylls, lycopene, β-carotene, and TCC (*r* = 0.95, 0.85, 0.95, and 0.90). In EVOO_DTL and EVOO_DTS, C* positively correlated only with TCC (*r* = 0.71 and 0.55) and lycopene content (*r* = 1.00).

The overall change of color during 12 months of storage was assessed by the ∆E*ab calculation (Figure 1). This color parameter was used to characterize the variation of colors in foods during processing. Data evidenced significant differences between ∆E*ab values in all samples (*p* < 0.05) (Appendix A). From eight months of storage, the ∆E*ab parameter for EVOO_DTD increased to reach 15.03 after 12 months storage.

EVOO phenolic compounds are responsible for the pungent taste as well as for the resistance to the oxidative process that reduces EVOO quality. Significant differences were observed in EVOO, EVOO_DTL, and DTD samples regarding TPC and TFC samples. EVOO_DTD showed values ranging from 110.80 to 142.24 mg CAE/g for TPC, and from 56.40 to 64.89 mg QE/g for TFC (Table 6). TPC values in the range of 110.80–90.21 mg CAE/g and TFC values in the range of 56.40 and 25.19 μg QE/g were observed in EVOO_DTL. Less content was found in EVOO ranging from 110.80 to 77.13 mg CAE/g for TPC and 56.40 to 22.31 mg QE/g for TFC.

Conversely, the enrichment of EVOO_DTD in TPC and TFC (31% and 8%, respectively) was related to the reduction of these phytochemicals in DTD (−13% and −20% for TPC and TFC, respectively) (Table 7). In comparison to EVOO, EVOO_DTL showed an initial increase in TPC and TFC (+7 and +2%, respectively). This enrichment is surely due to the release of these phytochemicals by the dried preserved tomatoes. Subsequently, a progressive reduction in these components was recorded in the tomato matrix.

De Abreu et al. [35] reported the TPC in the range from 337.8 to 835.7 mg GAE/100 g DM in eight preserved (in oil) dried tomatoes, whereas lower values were found by Toor and Savage [36] in semi-dehydrated tomatoes with a TPC ranging from 44.6 to 74.3 mg/100 g. This observation can be justified because the dried tomatoes are subjected to thermal treatment, which favors the liberation of phenolic compounds and lycopene from the cellular matrix. For this reason, lycopene and phenolic compounds are more bioavailable in dried tomatoes compared to semi-dehydrated with a consequent improvement of its functional and nutritional properties [33].

The determination of the presence of the microorganism in food above a level is considered an indication that food is produced and could be consumed under safe conditions. For these reasons, at the end of storage, all samples were subjected to microbiological investigation of mesophilic aerobic bacteria, yeast and moulds. All samples presented a lower bacterial population <10^2^ CFU\g as well as the growth of yeast and moulds at undetectable levels (<10^1^) (data not shown).

### 3.3. Fatty Acid Evolution in Dried Tomatoes Preserved in Oil

Table 8 reports the percentage fatty acid (FA) composition of all investigated EVOOs. During 12 months’ storage, fatty acids undergo significant modification. In all EVOOs, oleic acid represented the most abundant followed by linoleic acid and palmitoleic acid, but as expected, there were differences according to the storage conditions. Compared to EVOO_DTD and EVOO, in EVOO_DTL a more significant reduction was observed in palmitoleic (from 4.38 to 1.02%), oleic (from 79.92 to 53.47%), linoleic (from 7.10 to 0.71%) and α-linolenic acid (present in traces after the ninth month) due to the effects of light on the product. On the contrary, it is interesting to note that the content of stearic, arachidic, and gadoleic acids varies during storage. In fact, an initial increase was observed, followed by a progressive reduction, with more significant increases for samples stored in dark conditions. Indeed, in EVOO_DTL and EVOO_DTD, an initial increase in stearic acid (+2.2 for both samples), arachidic acid (+0.5%, and 0.7%, respectively), and gadoleic acid (+0.3%, and 0.6%, respectively) was observed, due to the contribution provided by the dried tomatoes.

The degradation of these fatty acids was less intense than in EVOO, due to the protective action exerted by the high tomato carotenoids and polyphenols content.

### 3.4. Antioxidant Activity

The determination of antioxidant capacity is an essential parameter for the shelf-life evaluation of foods. Several antioxidants may act in vivo through different mechanisms, and consequently no single test can fully define the antioxidant potential of a matrix.

For this reason, the goal of this work was to screen the antioxidant ability of samples using three methods: 2,2-azinobis (3-ethylbenzothiazoline-6-sulfonic acid) (ABTS), 1,1-diphenyl-2-picrylhydrazyl (DPPH), and β-carotene bleaching tests. This scientific evidence justifies the different results obtained in relation to the applied test. The radical scavenging activity of EVOOs and dried preserved tomatoes slices was examined using the DPPH radical and ABTS radical cation that possess a diverse stereochemistry and different training mechanisms.

The potential of samples to inhibit lipid peroxidation was assessed using the β-carotene bleaching test. A concentration-dependent activity was observed for all samples (Table 9). Interestingly, the antioxidant potential gradually increases in both EVOO_DTL and EVOO_DTD during storage compared to EVOO. This evidence may be explained by the enrichment of the oil with bioactive compounds from tomatoes. During the period of observation (12 months), a reduction in EVOO radical scavenging activity was observed for DPPH and ABTS tests with IC_50_ values from 90.17 to 110.69 μg/mL and from 85.43 to 98.46 μg/mL, respectively. A similar consideration was made for β-carotene bleaching tests, in which a reduction in lipid peroxidation protection was found with IC_50_ variations from 105.81 to 112.89 μg/mL and from 110.25 to 119.83 μg/mL, after 30 and 60 min of incubation, respectively.

Conversely, in EVOO_DTD, the antioxidant potential calculated as the percentage on IC_50_ value data increases by +43, +47, +48, and +38%, for the DPPH, ABTS, and β-carotene bleaching test after 30 and 60 min of incubation, respectively in comparison to EVOO alone. For EVOO_DTL, an increase of +30, 34, +32, +35% was found. Additionally, EVOO_DTD presented an antioxidant activity 2.4, 2.6, and 1.95 times higher in the DPPH, ABTS, and β-carotene bleaching test (30 min) in comparison to EVOO, respectively. Similarly, EVOO_DTD was 1.3, 1.4, and 1.3 times more active compared to EVOO_DTL in the DPPH, ABTS tests, and β-carotene bleaching test, respectively. These results demonstrated that tomatoes preserved and increased the antioxidant capacity of the EVOO_DTL and EVOO_DTD samples.

Interesting correlations were observed using Pearson’s statistical analysis. From data analysis, *r* values of 0.98, 0.97, 0.90 and 0.92 were found between the oleic acid (OA)/linoleic acid (LA) ratio and the DPPH, ABTS tests, and β-carotene bleaching test after 30 and 60 min, respectively.

Values of *r* of 0.97, 0.97, 0.90, and 0.93 were found between the MUFA/PUFA ratio and the DPPH, ABTS test, and β-carotene bleaching test after both incubation times. In EVOO_DTL, *r* values > 0.90 were observed when the MUFA or PUFA content was correlated with the DPPH, ABTS tests, and β-carotene bleaching test after 30 and 60 min.

In addition, the Pearson’s correlation coefficient was positive between TFC and the DPPH, ABTS tests, and β-carotene bleaching test after 30 and 60 min, with *r* = 0.78, 0.81, 0.74, and 0.81, respectively. A similar positive correlation between the MUFA and PUFA content and the antioxidant assay was observed in EVOO_DTD.

As expected, dried tomato samples showed a reduction in antioxidant activity with a particular decrease in DTD (Table 10). Already from the first month, there was a reduction of 10%, 8%, 4%, and 17% for DTL up to a 23%, 22%, 17%, and 30% decrease in the twelfth month, respectively, in the DPPH, ABTS, and β-carotene bleaching tests after 30 and 60 min of incubation. Indeed, IC_50_ variations in the range 35.23–58.44, 23.84–46.23, 55.68–72.57, and 58.37–88.54 μg/mL were found for the DPPH, ABTS tests, and β-carotene bleaching test after 30 and 60 min of incubation, respectively.

A more significant decrease was observed in both DTD and DTL samples but with main impact on DTD with IC_50_ values of 58.44, 46.23, 72.57, and 88.51 μg/mL (after 12 months) and 35.23, 23.84, 55.68, and 58.37 μg/mL (at 0 months storage) in the DPPH, ABTS, and β-carotene bleaching tests after 30 and 60 min of incubation, respectively. The radical scavenging activity decreased by −13% in both tests, while a reduction in lipid peroxidation protection of −16% and −24% after 30 and 60 min incubation, respectively, was observed. Interestingly, at the beginning of the observation period, both dried tomatoes had the greatest antioxidant activity compared to all EVOOs. After 12 months of storage, EVOO_DTD presented the greatest activity. Except for the radical scavenging activity, EVOO_DTL displayed an important protection of lipid peroxidation. In contrast, EVOO exhibited less antioxidant activity in all assays.

The RACI values of EVOOs and dried “San Marzano” tomato samples were calculated separately as the mean of standard scores transformed from the raw data generated with diverse antioxidant methods. Therefore, RACI was used to extrapolate samples with the highest antioxidant potential. Based on RACI data, the following antioxidant rank order was found: EVOO_DTD > EVOO_DTL > EVOO in each month (Appendix A).

Concerning dried tomatoes, DTD presented the highest antioxidant potential in each month of observation.

Just few studies investigated the antioxidant activity of dried tomatoes. Arslan et al. [40] investigated the effect of different drying processes (sun, oven, and microwave oven) on the antioxidant activity of tomato slices. A significant variability was observed in samples treated with different drying conditions. In particular, microwave oven drying at 700 W resulted in the most active effect independently by the applied antioxidant test. The application of osmotic pre-treatment with 10% saline solution to tomato slices before the oven drying process at 105 °C, for different times of exposure from 60 to 300 min, resulted in an efficient methodology to preserve bioactive compounds with DPPH radical scavenging potential [41]. More recently, Al Maiman et al. [42] demonstrated that storage of dried tomato slices resulted in the increase in antioxidant activity measured as radical scavenging potential (2.71 mg trolox/g DM for fresh tomato slice vs. 3.55 mg trolox/g DM for fresh tomato slice). Additionally, a similar finding was reported by Martinez-Valverde et al.’s [43] method. De Abreu et al. [35] analysed the antioxidant activity of hydrophilic and hydrophobic extracts from eight preserved (in oil) dried tomatoes. The radical scavenging activity was investigated using the DPPH test at a concentration 1.25 mg/mL, finding that the hydrophilic extracts had a greater capacity to sequester the DPPH radical than the hydrophobic ones (ranging 16–39.15% vs. 13.13–15.56%). On the contrary, in the case of the β-carotene/linoleic acid system, the hydrophobic extracts (from 72.5 to 86.65%) had a higher antioxidant capacity than the hydrophilic ones (from 72.5 to 86.65% vs. <22 to >60%). Furthermore, it should be considered that the total antioxidant activity of tomatoes is the result of the synergistic effect of the different bioactive compounds contained in the tomato matrix (carotenoids, phenols, and vitamins).

### 3.5. Principal Component Analysis

Data were analysed by means of PCA to determine the systematic variation and underlying relationships between bioactive compounds in the samples and antioxidant properties. All the determinations previously described were used to develop the PCA model. The first two PCs explained 91.3% of the variance in the data (PC1 = 60.37 and PC2 = 30.93), which was high enough to represent all the variables. The score plot for PC1 versus PC2 clearly distinguished three groups defined by length of storage, indicating that storage time had a major influence on the quality.

Figure 2a shows the PCA data concerning the content of active compounds in the samples and their antioxidant activity. The first component mainly correlated with the chlorophylls-EVOO, carotenoids-EVOO, β-carotene-EVO, chlorophylls-EVOO-DTL, β-carotene-DTL, chlorophylls-EVOO-DTD, lycopene-DTL, β-carotene-DTL, carotenoids-DTL, lycopene-DTD, β-carotene-DTD, carotenoids-DTD, and TPC-EVOO. It was negatively correlated with the acidity-EVOO, NP-EVOO, ΔK-EVOO, acidity-EVOO-DTL, NP-EVOO-DTL, ΔK-EVOO-DTL, acidity-EVOO-DTD, NP-EVOO-DTD, and ΔK-EVOO-DTD.

The second principal component correlated with the acidity-EVOO, NP-EVOO, ΔK-EVOO, acidity-EVOO-DTL, NP-EVOO-DTL, ΔK-EVOO-DTL, acidity-EVOO-DTD, NP-EVOO-DTD, ΔK-EVOO-DTD, acidity-EVOO-DTD, NP-EVOO-DTD, ΔK-E-DTD, carotenoids-EVOO-DTD, and lycopene-EVOO-DTD, while it negatively correlated with chlorophylls-EVOO, carotenoids-EVOO, β-carotene-EVOO, chlorophylls-EVOO-DTL, lycopene-DTL, β-carotene-DTL, carotenoids-DTL, lycopene-DTD, β-carotene-DTD, carotenoids-DTD, and TPC-EVOO. Two major groups based on the storage time were observed (Figure 2a).

Samples, which lie close to each other, are similar, while those away from the origin are extreme samples. Apart from the regular groupings of all storage, in intervals there were certain samples lying outside of these clusters (T2, T3, T4, T5, T6, and T7); moreover, they were negatively correlated to PC1. The control samples (T0) and (T1) exhibited a completely different behavior, showing extreme values and not a part of any cluster. The samples DTL and EVOO-DTL with a high content of carotenoids and β-carotene belonging to the first and second sampling (T0 and T1) were mostly located above the *x* axis. Principal component analysis of the fatty acids percentage gives two linear combinations which explain, overall, 90.64% of the variance, in particular 67.78% for the first component and 22.86% for the second.

Figure 2b shows the vectors of each variable and the distribution of the oil samples in the plane defined by the values of the two principal components. The first component was mainly correlated with C16:0, C16:1, C17:1, C18:0, C18:1, C18:2, C20:0, C18:3, C20:1, OA/LA, SFA, and MUFA for EVO, EVOO-DTL, and EVOO-DTD, while it correlates negatively with MUFA/PUFA for EVOO, EVOO-DTL, and EVOO-DTD. The second component was correlated positively with C14:0 and C17:0 for EVOO, EVOO-DTL, and EVOO_DTD. Three groups can be observed based on the similarities between the samples (T0-T1, T2-T10, and T11-T12). In each group, as shown in the Figure 2b, the samples are very similar to each other.

## 4. Conclusions

In Mediterranean countries, various vegetables are preserved in Extra Virgin Olive Oil (EVOO) to be consumed even outside their harvesting period.

“San Marzano” DPO dried tomato slices preserved in EVOO are a traditional Italian product. This is the first work that analysed the evolution of shelf life in dried tomato slices stored in EVOO for 12 months. The analysis of the data shows how storage in the dark prevents the degradation processes of this product.

In particular, after six months of storage, EVOO, used as a preserving liquid, is enriched by the phytochemicals present in dried tomatoes. The contribution of these phytochemicals may delay the degradation of the EVOO as confirmed by Principal component analysis, which showed how chlorophylls, carotenoids, lycopene, β-carotene, TPC, acidity, NP, and ΔK variables were most involved in the explained total variance. The score plot for PC1 versus PC2 clearly distinguished three groups indicating that shelf life significantly affects product quality.

## Figures and Tables

**Figure 1 foods-10-01706-f001:**
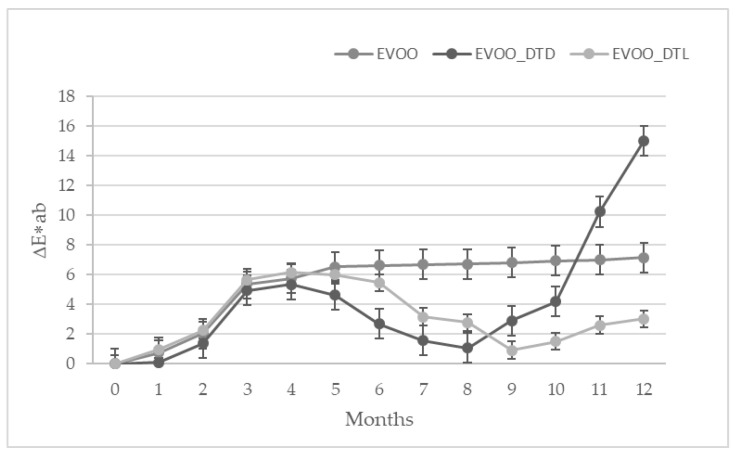
Evolution of ∆E*ab values during 12 months of storage.

**Figure 2 foods-10-01706-f002:**
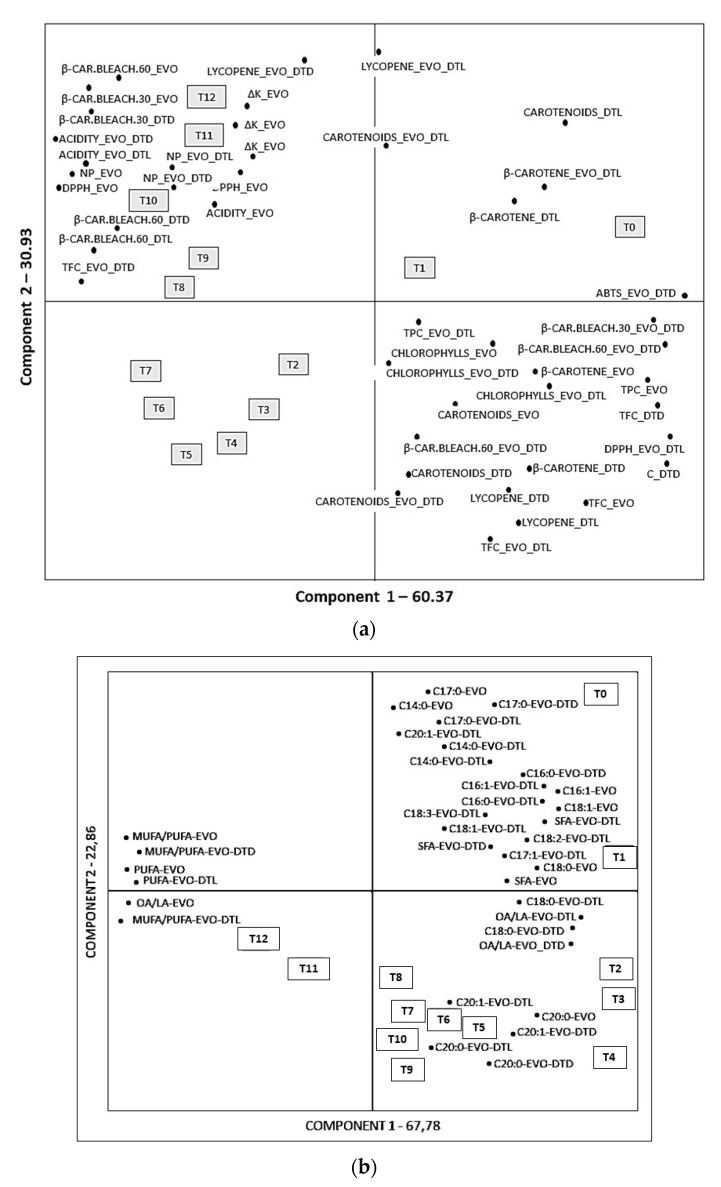
(**a**) Score plot of principal components 1 and 2 for chemical parameters, bioactive molecules, and an-tioxidant activity of the tomato dried and extra virgin olive oil samples stored in light (DTL) and darkness (DTD). (**b**) Score plot of principal components 1 and 2 for fatty acids of the tomato dried and extra virgin ol-ive oil stored in light (DTL) and darkness (DTD). T0-T12: months of analytical monitoring.

**Table 1 foods-10-01706-t001:** Acronyms of analysed samples.

Acronyms	Details
EVOO	Extra virgin olive oil
DTD	Dried tomato storage in dark condition
DTL	Dried tomato storage in light condition
EVOO_DTD	Dried tomato preserved in extra virgin olive oil storage in dark condition
EVOO_DTL	Dried tomato preserved in extra virgin olive oil storage in light condition

**Table 2 foods-10-01706-t002:** Chemical-physical quality parameters of EVOO, EVOO_DTL, and EVOO_DTD samples.

Months	EVOO			EVOO_DTL			EVOO_DTD					
	^^^ Acidity (%)	^°^ PN	^#^ ∆K	^^^ Acidity (%)	^°^ PN	^#^ ∆K	^^^ Acidity (%)	^°^ PN	^#^ ∆K	^^^ Sign.	^°^ Sign.	^#^ Sign.
0	0.68 ± 0.07 ^hB^	15.55 ± 1.54 ^nA^	0.01 ± 0.00 ^cC^	0.68 ± 0.07 ^hB^	15.55 ± 1.54 ^mA^	0.01 ± 0.00 ^cC^	0.68 ± 0.07 ^hB^	15.55 ± 1.53 ^mA^	0.01 ± 0.00 ^cC^	ns	ns	ns
1	0.68 ± 0.06 ^hC^	15.65 ± 1.58 ^mB^	0.01 ± 0.00 ^cE^	0.69 ± 0.06 ^hC^	15.84 ± 1.58 ^lA^	0.01 ± 0.00 ^cE^	0.64 ± 0.06 ^iD^	15.66 ± 1. 56 ^lB^	0.01 ± 0.00 ^cE^	*	**	ns
2	0.68 ± 0.06 ^hC^	16.66 ± 1.63 ^lA^	0.01 ± 0.00 ^cD^	0.70 ± 0.07 ^hC^	16.70 ± 1.61 ^iB^	0.01 ± 0.00 ^cD^	0.66 ± 0.06 ^iC^	16.39 ± 1.61 ^iB^	0.01 ± 0.00 ^cD^	ns	**	ns
3	0.74 ± 0.07 ^gD^	16.80 ± 1.68 ^iB^	0.01 ± 0.00 ^cE^	0.74 ± 0.07 ^gD^	17.05 ± 1.69 ^iA^	0.01 ± 0.00 ^cE^	0.69 ± 0.07 ^gD^	16.60 ± 1.64 ^hB^	0.01 ± 0.00 ^Ce^	ns	**	ns
4	0.78 ± 0.08 ^fC^	17.76 ± 1.71 ^hA^	0.01 ± 0.00 ^cF^	0.75 ± 0.08 ^gD^	17.75 ± 1.75 ^hA^	0.01 ± 0.00 ^cF^	0.71 ± 0.07 ^fE^	17.04 ± 1.72 ^gB^	0.01 ± 0.00 ^cF^	**	**	ns
5	0.82 ± 0.08 ^eD^	18.17 ± 1.83 ^gB^	0.01 ± 0.00 ^cG^	0.77 ± 0.08 ^fE^	18.25 ± 1.83 ^gA^	0.01 ± 0.00 ^cG^	0.72 ± 0.07 ^fF^	17.16 ± 1.74 ^gC^	0.01 ± 0.00 ^cG^	**	**	ns
6	0.86 ± 0.08 ^dD^	19.56 ± 1.91 ^fA^	0.01 ± 0.00 ^cG^	0.81 ± 0.08 ^eE^	19.18 ± 1.94 ^fB^	0.01 ± 0.00 ^cG^	0.76 ± 0.07 ^eF^	17.69 ± 1.78 ^fC^	0.01 ± 0.00 ^cG^	**	**	ns
7	0.89 ± 0.08 ^cD^	19.76 ± 1.96 ^eA^	0.01 ± 0.00 ^cG^	0.85 ± 0.08 ^E^	19.56 ± 1.98 ^eB^	0.01 ± 0.00 ^cG^	0.80 ± 0.08 ^efF^	17.75 ± 1.80 ^eC^	0.01 ± 0.00 ^cG^	**	**	ns
8	0.91 ± 0.08 ^bcD^	19.76 ± 1.97 ^eA^	0.17 ± 0.01 ^bG^	0.86 ± 0.09 ^dE^	19.57 ± 1.99 ^dB^	0.15 ± 0.01 ^bH^	0.83 ± 0.08 ^dF^	17.76 ± 1.80 ^eC^	0.14 ± 0.01 ^bH^	**	**	ns
9	0.93 ± 0.09 ^bD^	22.65 ± 2.14 ^dA^	0.17 ± 0.01 ^bF^	0.91 ± 0.09 ^cD^	20.24 ± 2.01 ^cB^	0.15 ± 0.01 ^bG^	0.87 ± 0.08 ^cE^	18.76 ± 1.81 ^dC^	0.14 ± 0.01 ^bG^	**	**	*
10	0.96 ± 0.09 ^aD^	23.62 ± 2.20 ^cA^	0.17 ± 0.01 ^bF^	0.95 ± 0.15 ^cD^	20.75 ± 2.04 ^cB^	0.16 ± 0.02 ^bF^	0.89 ± 0.09 ^cE^	18.97 ± 1.83 ^cC^	0.15 ± 0.01 ^aF^	**	**	ns
11	0.96 ± 0.09 ^aD^	25.18 ± 2.51 ^bA^	0.39 ± 0.03 ^aF^	0.96 ± 0.16 ^bD^	21.53 ± 2.10 ^bB^	0.16 ± 0.02 ^aF^	0.91 ± 0.12 ^bE^	19.17 ± 1.91 ^bC^	0.15 ± 0.01 ^aF^	**	**	ns
12	0.98 ± 0.12 ^aD^	26.24 ± 2.63 ^aA^	0.40 ± 0.05 ^aF^	0.97 ± 0.18 ^aD^	22.54 ± 2.19 ^aB^	0.17 ± 0.02 ^aG^	0.94 ± 0.14 ^aE^	20.71 ± 2.02 ^aC^	0.16 ± 0.02 ^aG^	**	**	**
Sign.	**	**	**	**	**	**	**	**	**			

Data are reported as mean ± standard deviation (*n* = 3). PN: Peroxides number; differences were evaluated by one-way analysis of variance (ANOVA) test completed with a multicomparison Tukey’s test. Means in the same row with different capital letters differ significantly (*p* < 0.05); means in the same column with different small letters differ significantly. ns: not significance; Significance at * *p* < 0.05, ** *p* < 0.01.

**Table 3 foods-10-01706-t003:** β-Carotene and lycopene content in EVOO, EVOO_DTL, and EVOO_DTD samples (mg/100 g).

Months	EVOO	EVOO_DTL		EVOO_DTD			
	^^^ β-carotene	^#^ Lycopene	^^^ β-carotene	^#^ Lycopene	^^^ β-carotene	^^^ Sign.	^#^ Sign.
0	90.26 ± 3.32 ^aA^	0.91 ± 0.02 ^cB^	90.26 ± 4.23 ^aA^	0.91 ± 0.02 ^aB^	90.26 ± 5.36 ^aA^	ns	ns
1	89.04 ± 3.56 ^bB^	0.84 ± 0.03 ^dC^	89.1 ± 4.02 ^bB^	0.90 ± 0.03 ^bC^	90.71 ± 5.02 ^aA^	ns	*
2	77.36 ± 4.21 ^cB^	0.79 ± 0.01 ^eD^	68.32 ± 4.55 ^hC^	0.85 ± 0.03 ^hD^	87.29 ± 4.89 ^bA^	ns	**
3	66.52 ± 3.23 ^dB^	0.67 ± 0.01 ^gD^	66.33 ± 3.26 ^iC^	0.70 ± 0.02 ^iD^	87.00 ± 4.61 ^bA^	ns	**
4	61.84 ± 5.02 ^eC^	0.63 ± 0.02 ^hD^	65.57 ± 3.56 ^lB^	0.69 ± 0.04 ^iD^	78.23 ± 4.36 ^lA^	ns	**
5	58.51 ± 3.12 ^fC^	0.64 ± 0.02 ^hD^	66.14 ± 2.859 ^iB^	0.73 ± 0.04 ^lD^	78.67 ± 3.89 ^iA^	ns	**
6	48.37 ± 5.03 ^gC^	0.66 ± 0.02 ^hE^	68.36 ± 5.02 ^hB^	0.78 ± 0.05 ^hD^	79.40 ± 3.84 ^hA^	*	**
7	45.42 ± 4.23 ^hC^	0.74 ± 0.05 ^fD^	73.36 ± 5.15 ^gB^	0.84 ± 0.05 ^gD^	81.80 ± 4.05 ^gA^	ns	**
8	44.32 ± 4.06 ^iC^	0.75 ± 0.03 ^fE^	75.71 ± 4.25 ^dB^	0.96 ± 0.03 ^dD^	82.90 ± 4.08 ^fA^	*	**
9	42.66 ± 2.89 ^lC^	0.84 ± 0.04 ^eE^	75.96 ± 4.46 ^cB^	1.04 ± 0.02 ^cD^	83.70 ± 4.11 ^eA^	*	**
10	42.63 ± 3.89 ^mC^	0.95 ± 0.04 ^cD^	75.01 ± 3.78 ^eB^	1.10 ± 0.03 ^eD^	84.67 ± 3.55 ^dA^	ns	**
11	41.33 ± 3.54 ^nC^	1.00 ± 0.02 ^bE^	74.71 ± 3.45 ^fB^	1.38 ± 0.03 ^fD^	85.79 ± 3.26 ^dA^	*	**
12	38.61 ± 3.66 ^oB^	1.10 ± 0.02 ^aC^	73.28 ± 3.02 ^gA^	1.59 ± 0.06 ^gC^	85.90 ± 5.02 ^cA^	ns	**
Sign.	**	**	**	**	**		

Data are reported as mean ± standard deviation (*n* = 3). Differences were evaluated by one-way analysis of variance (ANOVA) test completed with a multicomparison Tukey’s test. Means in the same row with different capital letters differ significantly (*p* < 0.05); means in the same column with different small letters differ significantly (*p* < 0.05). ns: not significance; * Significance at *p* < 0.05, ** *p* < 0.01.

**Table 4 foods-10-01706-t004:** β-carotene, lycopene, and TCC in “San Marzano” DTL and DTD samples (mg/100 g).

Months	DTL			DTD					
	^ Lycopene	° β-carotene	# TCC	^ Lycopene	° β-carotene	# TCC	^ Sign.	° Sign.	# Sign.
0	0.98 ± 0.03 ^aC^	17.05 ± 1.02 ^aB^	24.57 ± 2.02 ^aA^	0.98 ± 0.02 ^aC^	17.05 ± 2.30 ^aB^	24.57 ± 3.12 ^aA^	ns	ns	ns
1	0.97 ± 0.02 ^aE^	16.18 ± 1.11 ^bD^	24.07 ± 2.23 ^bB^	0.97 ± 0.03 ^abE^	16.43 ± 1.89 ^bC^	24.12 ± 2.56 ^bA^	ns	*	*
2	0.97 ± 0.03 ^aC^	15.92 ± 0.98 ^cB^	23.73 ± 2.55 ^cA^	0.96 ± 0.03 ^abC^	15.92 ± 2.02 ^dB^	23.74 ± 2.87 ^cA^	ns	ns	ns
3	0.97 ± 0.01 ^abE^	15.9 ± 0.86 ^cdC^	23.61 ± 1.56 ^dB^	0.96 ± 0.01 ^abE^	15.38 ± 2.33 ^fD^	23.67 ± 1.19 ^dA^	ns	**	*
4	0.95 ± 0.01 ^bE^	15.9 ± 1.03 ^cdC^	22.43 ± 1.87 ^gB^	0.94 ± 0.02 ^bE^	15.81 ± 2.56 ^eD^	23.14 ± 1.55 ^eA^	ns	**	**
5	0.93 ± 0.04 ^bE^	15.88 ± 1055 ^dD^	22.6 ± 2.05 ^eB^	0.90 ± 0.02 ^cE^	16.01 ± 1.84 ^cC^	22.94 ± 2.09 ^fA^	ns	**	**
6	0.92 ± 0.03 ^cE^	15.23 ± 2.03 ^fC^	22.54 ± 2.86 ^fA^	0.90 ± 0.03 ^cE^	15.06 ± 1.56 ^gD^	22.07 ± 2.37 ^gB^	ns	**	**
7	0.88 ± 004 ^cE^	14.87 ± 2.01 ^gC^	21.61 ± 1.56 ^hA^	0.88 ± 0.03 ^cE^	14.24 ± 1.84 ^hD^	21.54 ± 2.22 ^hB^	ns	**	**
8	0.83 ± 0.03 ^dF^	13.67 ± 1.02 ^hC^	20.64 ± 1.84 ^iB^	0.87 ± 0.04 ^cE^	13.26 ± 2.55 ^iD^	21.03 ± 1.84 ^iA^	*	**	**
9	0.78 ± 0.00 ^eE^	15.36 ± 1.05 ^eC^	20.01 ± 1.91 ^lB^	0.54 ± 0.03 ^dF^	12.98 ± 2.36 ^lD^	20.82 ± 2.64 ^lA^	**	**	**
10	0.67 ± 0.02 ^fE^	13.02 ± 0.93 ^iC^	19.84 ± 2.63 ^mB^	0.54 ± 0.03 ^dF^	12.31 ± 2.34 ^mD^	20.52 ± 1.05 ^mA^	**	**	**
11	0.68 ± 0.02 ^fE^	12.47 ± 0.88 ^lC^	19.02 ± 2.84 ^nB^	0.51 ± 0.02 ^deF^	12.01 ± 1.55 ^nD^	20.02 ± 1.55 ^nA^	**	**	**
12	0.67 ± 0.03 ^fE^	12.04 ± 1.25 ^mC^	18.74 ± 2.22 ^oB^	0.49 ± 0.01 ^eF^	11.90 ± 1.27 ^oD^	19.96 ± 1.23 ^oA^	**	**	**
Sign.	**	**	**	**	**	**			

Data are reported as mean ± standard deviation (*n* = 3). TCC: total carotenoid content; differences were evaluated by one-way analysis of variance (ANOVA) test completed with a multicomparison Tukey’s test. Means in the same row with different capital letters differ significantly (*p* < 0.05); means in the same column with different small letters differ significantly (*p* < 0.05). ns: not significance; * *p* < 0.05, ** *p* < 0.01.

**Table 5 foods-10-01706-t005:** CIELab color parameters (L*, C*, and H*) in EVOO, EVOO_DTD, and EVOO_DTL.

Months	EVOO			EVOO_DTL			EVOO_DTD					
	^^^ L*	^°^ C*	^#^ H*	^^^ L*	^°^ C*	^#^ H*	L*	^°^ C*	^#^ H*	^^^ Sign.	^°^ Sign.	^#^ Sign.
0	29.45 ± 0.82 ^hB^	19.03 ± 1.22 ^aC^	86.80 ± 3.03 ^bA^	29.45 ± 1.88 ^hB^	19.03 ± 2.02 ^dC^	86.80 ± 3.03 ^aA^	29.45 ± 1.22 ^fB^	19.03 ± 2.03 ^fC^	86.80 ± 3.02 ^bA^	ns	ns	ns
1	29.89 ± 0.69 ^gC^	18.37 ± 1.02 ^bG^	86.81 ± 3.22 ^bA^	29.97 ± 1.56 ^fB^	18.16 ± 1.88 ^eH^	86.83 ± 3.12 ^aA^	29.53 ± 1.03 ^eD^	18.79 ± 1.88 ^gF^	86.86 ± 2.88 ^aA^	**	**	ns
2	30.11 ± 1.02 ^fD^	17.11 ± 1.11 ^cG^	87.35 ± 2.88 ^aA^	30.23 ± 2.02 ^eC^	16.9 ± 1.45 ^fH^	86.83 ± 0.24 ^aB^	30.04 ± 1.45 ^dE^	17.74 ± 1.96 ^hF^	86.83 ± 2.63 ^abB^	**	**	**
3	31.97 ± 1.03 ^eB^	14.23 ± 1.03 ^dF^	86.81 ± 2.45 ^bA^	32.05 ± 2.03 ^cB^	13.98 ± 1.23 ^lG^	86.80 ± 2.88 ^aA^	31.69 ± 2.02 ^bC^	16.61 ± 2.06 ^lD^	86.81 ± 2.12 ^bA^	*	**	ns
4	32.54 ± 1.11 ^dD^	14.14 ± 1.06 ^eG^	86.83 ± 2.65 ^bA^	32.71 ± 1.88 ^aC^	13.78 ± 1.11 ^mH^	86.79 ± 2.96 ^aB^	32.15 ± 2.03 ^aE^	14.4 ± 1.45 ^oF^	86.82 ± 2.10 ^bAB^	**	**	**
5	32.58 ± 0.99 ^dD^	13.27 ± 1.04 ^fH^	86.8 ± 3.02 ^bA^	32.73 ± 1.45 ^aC^	13.98 ± 1.12 ^lG^	80.80 ± 2.45 ^cB^	32.16 ± 2.11 ^aE^	15.24 ± 1.62 ^nF^	86.80 ± 2.24 ^bA^	**	**	**
6	32.64 ± 0.89 ^cC^	13.21 ± 1.12 ^fH^	86.82 ± 3.05 ^bA^	32.41 ± 2.36 ^bD^	14.40 ± 1.14 ^iG^	80.81 ± 2.85 ^cB^	30.41 ± 1.88 ^cE^	16.49 ± 1.69 ^mF^	86.83 ± 2.02 ^abA^	**	**	**
7	32.67 ± 0.88 ^cC^	13.15 ± 1.13 ^gH^	86.81 ± 2.88 ^bA^	30.86 ± 2.45 ^dD^	16.17 ± 1.12 ^hG^	83.80 ± 3.06 ^bB^	30.01 ± 1.75 ^dE^	17.53 ± 1.85 ^iF^	86.83 ± 2.21 ^aA^	**	**	*
8	32.60 ± 1.02 ^cB^	13.06 ± 1.02 ^hG^	86.79 ± 2.63 ^bA^	30.01 ± 2.45 ^fC^	16.28 ± 1.02 ^gF^	86.79 ± 3.03 ^aA^	29.14 ± 1.62 ^gD^	20.04 ± 1.77 ^eE^	86.82 ± 2.11 ^bA^	**	**	ns
9	32.71 ± 1.12 ^cC^	13.01 ± 1.02 ^iH^	86.78 ± 2.24 ^bB^	29.82 ± 2.30 ^gD^	18.16 ± 1.03 ^eG^	86.81 ± 3.11 ^aA^	28.37 ± 2.45 ^hE^	21.71 ± 1.62 ^dF^	86.80 ± 2.03 ^bA^	**	**	ns
10	32.89 ± 1.10 ^bBC^	12.98 ± 0.99 ^lC^	80.82 ± 3.08 ^cAB^	29.01 ± 3.02 ^iC^	20.46 ± 1.02 ^cC^	86.81 ± 3.03 ^aA^	28.09 ± 2.66 ^iC^	22.96 ± 1.82 ^cC^	86.80 ± 1.77 ^bA^	ns	ns	*
11	33.01 ± 1.10 ^aC^	12.97 ± 1.06 ^lmH^	80.82 ± 3.11 ^cB^	28.72 ± 3.11 ^lD^	21.50 ± 2.02 ^bG^	86.82 ± 2.44 ^aA^	26.51 ± 1.55 ^lE^	28.81 ± 1.74 ^bF^	86.76 ± 1.88 ^cAB^	**	**	**
12	33.05 ± 1.05 ^aE^	12.82 ± 1.85 ^nL^	86.77 ± 2.45 ^bB^	27.54 ± 3.56 ^mF^	24.01 ± 2.33 ^aH^	86.8 ± 2.88 ^aA^	24.47 ± 1.23 ^mG^	33.19 ± 1.71 ^aC^	86.80 ± 3.03 ^bAB^	**	**	*
Sign.	**	**	**	**	**	**	**	**	**			

Data are reported as mean ± standard deviation (*n* = 3). Differences were evaluated by one-way analysis of variance (ANOVA) test completed with a multicomparison Tukey’s test. Means in the same row with different capital letters differ significantly (*p* < 0.05); means in the same column with different small letters differ significantly (*p* < 0.05). ns: not significance; * *p* < 0.05, ** *p* < 0.01.

**Table 6 foods-10-01706-t006:** TPC (mg CAE/g) and TFC (mg QE/g) content in EVOO, EVOO_DTD, and EVOO_DTL samples.

Months	EVOO		EVOO_DTL		EVOO_DTD			
	^^^ TPC	^#^ TFC	^^^ TPC	^#^ TFC	^^^ TPC	^#^ TFC	^^^ Sign.	^#^ Sign.
0	110.80 ± 3.51 ^aA^	56.40 ± 1.90 ^aC^	110.80 ± 3.52 ^bA^	56.40 ± 2.02 ^gC^	110.80 ± 3.52 ^aA^	56.40 ± 2.02 ^aC^	ns	ns
1	92.38 ± 3.02 ^bC^	55.58 ± 1.93 ^bF^	96.76 ± 2.92 ^gB^	56.81 ± 2.14 ^fE^	121.34 ± 3.90 ^nA^	57.31 ± 2.13 ^nE^	**	*
2	91.56 ± 2.90 ^cC^	53.17 ± 1.62 ^cG^	96.03 ± 2.91 ^hB^	57.01 ± 2.15 ^eF^	122.46 ± 3.82 ^mA^	58.65 ± 1.92 ^mE^	**	**
3	90.85 ± 2.92 ^dC^	52.85 ± 1.52 ^dH^	96.55 ± 3.16 ^gB^	57.36 ± 2.00 ^dG^	123.81 ± 3.83 ^lA^	60.76 ± 1.96 ^lF^	**	**
4	90.13 ± 2.82 ^eC^	50.96 ± 1.05 ^eG^	97.01 ± 3.73 ^fB^	57.80 ± 1.92 ^cF^	125.36 ± 4.06 ^iA^	61.58 ± 1.83 ^iE^	**	**
5	89.71 ± 2.60 ^fC^	50.12 ± 1.06 ^fH^	97.13 ± 3.82 ^fB^	58.31 ± 1.57 ^bG^	126.87 ± 4.13 ^hA^	61.99 ± 1.74 ^hF^	**	**
6	88.52 ± 2.44 ^gC^	46.31 ± 0.91 ^gG^	98.62 ± 3.50 ^dB^	58.94 ± 1.44 ^aF^	127.01 ± 4.13 ^gA^	62.67 ± 1.66 ^gE^	**	**
7	87.31 ± 2.42 ^hC^	41.99 ± 0.82 ^hG^	101.17 ± 3.44 ^cB^	54.39 ± 1.22 ^hF^	128.93 ± 4.36 ^fA^	62.96 ± 1.44 ^fE^	**	**
8	85.11 ± 2.32 ^iC^	37.41 ± 0.72 ^iG^	117.92 ± 3.23 ^aB^	51.24 ± 1.25 ^iF^	131.74 ± 4.44 ^eA^	63.15 ± 1.37 ^eE^	**	**
9	81.59 ± 2.22 ^lC^	35.11 ± 0.60 ^lG^	97.54 ± 2.94 ^eB^	40.61 ± 1.06 ^lF^	132.28 ± 4.46 ^dA^	63.83 ± 1.25 ^dE^	**	**
10	80.15 ± 2.21 ^mC^	27.43 ± 0.53 ^mG^	94.47 ± 2.87 ^iB^	34.56 ± 0.91 ^mF^	134.05 ± 4.53 ^cA^	64.11 ± 0.95 ^cE^	**	**
11	78.84 ± 2.10 ^nC^	25.92 ± 0.54 ^nG^	90.98 ± 2.52 ^lB^	33.89 ± 0.92 ^nF^	140.18 ± 4.66 ^bA^	64.72 ± 0.92 ^bF^	**	**
12	77.13 ± 2.13 ^oC^	22.31 ± 0.37 ^oI^	90.21 ± 2.56 ^mB^	25.19 ± 0.75 ^oG^	142.24 ± 4.81 ^aA^	64.89 ± 0.73 ^aF^	**	**
Sign.	**	**	**	**	**	**		

Data are reported as mean ± standard deviation (*n* = 3). TPC: total phenol content; TFC: total flavonoid content; differences were evaluated by one-way analysis of variance (ANOVA) test completed with a multicomparison Tukey’s test. Means in the same row with different capital letters differ significantly (*p* < 0.05); means in the same column with different small letters differ significantly (*p* < 0.05). ns: not significance; * *p* < 0.05, ** *p* < 0.01.

**Table 7 foods-10-01706-t007:** TPC (mg CAE/g) and TFC (mg QE/g) content in DTL and DTD samples.

Months	DTL		DTD			
	^^^ TPC	^#^ TFC	^^^ TPC	^#^ TFC	^^^ Sign.	^#^ Sign.
0	90.30 ± 3.01 ^aB^	42.30 ± 1.72 ^aD^	90.30 ± 3.01 ^aB^	42.30 ± 1.71 ^aD^	ns	ns
1	87.28 ± 2.92 ^bD^	36.32 ± 1.40 ^bH^	88.44 ± 2.42 ^bD^	38.87 ± 0.82 ^bG^	ns	**
2	86.43 ± 2.83 ^cD^	33.11 ±1.21 ^cI^	87.20 ± 2.90 ^cD^	37.16 ± 0.75 ^cH^	ns	**
3	83.11 ± 2.60 ^dE^	32.83 ± 1.25 ^dM^	86.95 ± 2.82 ^dD^	36.85 ± 0.76 ^dI^	**	**
4	81.37 ± 2.55 ^eD^	30.34 ± 1.13 ^eI^	82.12 ± 2.53 ^eD^	35.85 ± 0.62 ^eH^	ns	**
5	80.48 ± 2.48 ^fE^	28.58 ± 1.02 ^fM^	81.63 ± 2.50 ^fD^	33.33 ± 1.23 ^fI^	**	**
6	80.19 ± 2.42 ^gD^	27.49 ± 0.90 ^gI^	81.22 ± 2.51 ^gD^	31.31 ± 1.10 ^gH^	ns	**
7	79.42 ± 2.33 ^hD^	25.97 ± 0.92 ^hI^	80.83 ± 2.45 ^hD^	29.31 ± 1.07 ^hH^	ns	**
8	78.64 ± 2.37 ^iD^	25.26 ± 0.82 ^iI^	79.90 ± 2.20 ^iD^	26.12 ± 0.95 ^iH^	ns	*
9	78.15 ± 2.20 ^lD^	24.63 ± 0.73 ^lI^	79.62 ± 2.12 ^lD^	25.85 ± 0.84 ^lH^	*	*
10	77.69 ± 2.12 ^mD^	22.51 ± 0.54 ^mI^	78.95 ± 2.06 ^mD^	25.14 ± 0.82 ^mH^	ns	**
11	75.77 ± 2.01 ^nD^	22.11 ± 0.53 ^nI^	78.41 ± 1.95 ^nC^	23.13 ± 0.60 ^nH^	**	**
12	74.63 ± 2.02 ^oE^	21.72 ± 0.42 ^oM^	76.99 ± 1.80 ^oD^	22.52 ± 0.52 ^oI^	**	**
Sign.	**	**	**	**		

Data are reported as mean ± standard deviation (*n* = 3). Differences were evaluated by one-way analysis of variance (ANOVA) test completed with a multicomparison Tukey’s test. Means in the same row with different capital letters differ significantly (*p* < 0.05); means in the same column with different small letters differ significantly (*p* < 0.05). ns: not significance; * Significance at *p* < 0.05, ** Significance at *p* < 0.01.

**Table 8 foods-10-01706-t008:** Fatty acid profile of EVOO, EVOO_DTL, and EVOO_DTD samples (%).

	Months													Sign.
	0	1	2	3	4	5	6	7	8	9	10	11	12	
**EVOO**														
Myristic acid	0.02 ± 0.01 ^a^	0.01 ± 0.00 ^b^	tr	tr	tr	tr	tr	tr	tr	tr	tr	tr	tr	*
Palmitic acid	0.83 ± 0.02 ^a^	0.80 ± 0.02 ^a^	0.77 ± 0.03 ^a^	0.67 ± 0.02 ^a^	0.52 ± 0.02 ^ab^	0.47 ± 0.02 ^bc^	0.41 ± 0.02 ^cde^	0.38 ± 0.02 ^cde^	0.35 ± 0.02 ^cde^	0.21 ± 0.01 ^ef^	0.15 ± 0.02 ^f^	0.04 ± 0.01 ^de^	tr	**
Palmitoleic acid	4.38 ± 0.23 ^a^	4.28 ± 0.77 ^a^	3.74 ± 0.56 ^b^	3.17 ± 0.47 ^c^	2.8 ± 0.11 ^d^	2.75 ± 0.55 ^d^	2.69 ± 0.12 ^d^	2.31 ± 0.02 ^e^	2.10 ± 0.01 ^f^	2.07 ± 0.03 ^f^	2.07 ± 0.03 ^f^	2.05 ± 0.12 ^f^	1.02 ± 0.06 ^g^	**
Margaric acid	0.03 ± 0.01 ^a^	0.02 ± 0.01 ^a^	tr	tr	tr	tr	tr	tr	tr	tr	tr	tr	tr	ns
Margaroleic acid	0.76 ± 0.02 ^a^	0.74 ± 0.04 ^b^	0.70 ± 0.06 ^c^	0.67 ± 0.05 ^d^	0.6 ± 0.03 ^e^	0.54 ± 0.03 ^f^	0.50 ± 0.01 ^g^	0.41 ± 0.01 ^h^	0.38 ± 0.02 ^i^	0.31 ± 0.01 ^l^	0.3 ± 0.01 ^m^	0.25 ± 0.01 ^n^	tr	**
Stearic acid	3.65 ± 0.56 ^ab^	3.62 ± 0.56 ^ab^	3.59 ± 0.88 ^a^	3.46 ± 0.66 ^ab^	3.23 ± 0.25 ^b^	3.16 ± 0.25 ^b^	3.09 ± 0.14 ^b^	3.02 ± 0.11 ^b^	2.97 ± 0.23 ^b^	2.85 ± 0.25 ^b^	2.51 ± 0.12 ^b^	2.45 ± 0.25 ^b^	1.74 ± 0.08 ^c^	*
Oleic acid	79.92 ± 6.23 ^a^	78.05 ± 5.12 ^b^	75.34 ± 4.58 ^c^	72.1 ± 4.58 ^d^	71.08 ± 6.12 ^d^	70.87 ± 6.15 ^e^	70.68 ± 4.26 ^e^	69.64 ± 5.01 ^f^	69.55 ± 4.25 ^fg^	69.07 ± 4.89 ^g^	66.17 ± 3.69 ^hi^	64.23 ± 4.56	60.98 ± 4.02 ^l^	**
Linoleic acid	7.10 ± 0.87 ^a^	7.09 ± 0.88 ^b^	7.02 ± 0.99 ^b^	6.73 ± 0.87 ^c^	6.42 ± 0.93 ^d^	6.17 ± 0.44 ^e^	5.03 ± 0.36 ^f^	4.94 ± 0.12 ^g^	4.17 ± 0.54 ^h^	4.1 ± 0.67 ^i^	3.38 ± 0.12 ^l^	2.60 ± 0.21 ^m^	1.82 ± 0.08 ^n^	**
Arachidic acid	0.26 ± 0.02 ^f^	0.30 ± 0.02 ^e^	0.50 ± 0.02 ^a^	0.48 ± 0.02 ^a^	0.47 ± 0.02 ^a^	0.42 ± 0.02 ^b^	0.42 ± 0.02 ^b^	0.38 ± 0.01 ^c^	0.34 ± 0.02 ^d^	0.34 ± 0.02 ^d^	0.28 ± 0.02 ^ef^	0.21 ± 0.01 ^g^	0.14 ± 0.02 ^h^	**
α-Linolenic acid	0.83 ± 0.05 ^a^	0.81 ± 0.04 ^a^	0.76 ± 0.03 ^b^	0.7 ± 0.02 ^c^	0.62 ± 0.01 ^d^	0.58 ± 0.02 ^e^	0.49 ± 0.01 ^f^	0.44 ± 0.01 ^g^	0.32 ± 0.01 ^h^	0.21 ± 0.02 ^i^	0.12 ± 0.02 ^l^	tr	tr	**
Gadoleic acid	0.13 ± 0.02 ^a^	tr	tr	tr	tr	tr	tr	tr	tr	tr	tr	tr	tr	ns
OA/LA	11.26 ± 0.87 ^i^	11.01 ± 1.02 ^l^	10.73 ± 1.02 ^m^	10.71 ± 0.95 ^m^	11.07 ± 0.75 ^l^	11.49 ± 0.47 ^h^	14.05 ± 0.74 ^g^	14.10 ± 0.88 ^f^	16.68 ± 1.02 ^e^	16.85 ± 0.44 ^d^	19.58 ± 1.36 ^c^	24.70 ± 2.02 ^b^	33.51 ± 3.23 ^a^	**
∑ SFA	4.79 ± 0.36 ^ab^	4.75 ± 0.77 ^ab^	4.86 ± 0.88 ^ab^	4.61 ± 0.88 ^abc^	4.22 ± 0.63 ^abc^	4.05 ± 0.23 ^abc^	3.92 ± 0.12 ^a^	3.78 ± 0.45 ^abc^	3.66 ± 0.56 ^abc^	3.4 ± 0.12 ^abc^	2.94 ± 0.66 ^bc^	2.70 ± 0.89 ^bc^	1.88 ± 0.66 ^c^	**
∑ MUFA	84.43 ± 4.56 ^a^	82.33 ± 6.23 ^b^	79.08 ± 4.89 ^c^	75.27 ± 6.23 ^d^	73.88 ± 5.45 ^e^	73.62 ± 4.23 ^f^	73.37 ± 5.85 ^g^	71.95 ± 5.78 ^h^	71.65 ± 5.76 ^i^	71.14 ± 6.02 ^l^	68.24 ± 5.31 ^m^	66.28 ± 6.12 ^n^	62.12 ± 4.21 ^o^	**
∑ PUFA	7.93 ± 0.44 ^a^	7.90 ± 1.05 ^a^	7.78 ± 10.84 ^b^	7.43 ± 0.74 ^c^	7.04 ± 0.88 ^d^	6.75 ± 0.64 ^e^	5.52 ± 0.64 ^f^	5.38 ± 0.67 ^g^	4.49 ± 0.85 ^h^	4.31 ± 0.78 ^i^	3.5 ± 0.48 ^l^	2.60 ± 0.78 ^m^	1.82 ± 0.02 ^n^	**
MUFA/PUFA	10.65 ± 0.85 ^h^	10.42 ± 0.96 ^i^	10.16 ± 0.55 ^l^	10.13 ± 0.67 ^l^	10.49 ± 0.91 ^i^	10.91 ± 0.82 ^g^	13.29 ± 0.82 ^f^	13.37 ± 0.45 ^f^	15.96 ± 0.79 ^e^	16.51 ± 0.61 ^d^	19.50 ± 0.72 ^c^	25.49 ± 1.06 ^b^	34.07 ± 3.09 ^a^	**
**EVOO_DTL**														
Myristic acid	0.02 ± 0.01	tr	tr	tr	tr	tr	tr	tr	tr	tr	tr	tr	tr	ns
Palmitic acid	0.83 ± 0.01 ^a^	0.78 ± 0.02 ^a^	0.71 ± 0.02 ^b^	0.65 ± 0.03 ^c^	0.46 ± 0.02 ^d^	0.38 ± 0.01 ^e^	0.32 ± 0.01 ^f^	0.27 ± 0.01 ^fg^	0.22 ± 0.01 ^g^	0.15 ± 0.01 ^h^	0.02 ± 0.01 ^i^			
Palmitoleic acid	4.38 ± 0.12 ^a^	4.15 ± 0.14 ^b^	3.69 ± 0.14 ^c^	2.56 ± 0.04 ^d^	2.54 ± 0.63 ^d^	2.49 ± 0.36 ^e^	2.43 ± 0.21 ^f^	2.36 ± 0.22 ^g^	2.23 ± 0.36 ^h^	2.23 ± 0.25 ^h^	2.15 ± 0.85 ^i^	2.12 ± 0.45 ^l^	1.02 ± 0.03 ^m^	**
Margaric acid	0.03 ± 0.01 ^a^	0.02 ± 0.01 ^b^	tr	tr	tr	tr	tr	tr	tr	tr	tr	tr	tr	*
Margaroleic acid	0.76 ± 0.02 ^a^	0.73 ± 0.03 ^a^	0.64 ± 0.05 ^b^	0.58 ± 0.03 ^c^	0.56 ± 0.02 ^c^	0.55 ± 0.03 ^c^	0.51 ± 0.03 ^d^	0.47 ± 0.02 ^e^	0.39 ± 0.01 ^f^	0.34 ± 0.02 ^g^	0.32 ± 0.01 ^g^	0.23 ± 0.03 ^g^	tr	**
Stearic acid	3.65 ± 0.51 ^e^	3.82 ± 0.56 ^d^	4.19 ± 0.47 ^c^	4.89 ± 0.14 ^b^	5.85 ± 0.25 ^a^	3.73 ± 0.87 ^d^	2.67 ± 0.14 ^f^	2.57 ± 0.96 ^f^	2.41 ± 0.25 ^g^	2.25 ± 0.32 ^h^	2.19 ± 0.23 ^h^	1.53 ± 0.47 ^i^	0.17 ± 0.01 ^l^	**
Oleic acid	79.92 ± 5.36 ^a^	75.17 ± 4.23 ^b^	74.14 ± 5.23 ^c^	72.70 ± 4.23 ^d^	71.38 ± 4.56 ^e^	70.72 ± 4.56 ^f^	70.66 ± 4.28 ^f^	69.89 ± 4.23 ^g^	69.28 ± 3.54 ^g^	67.77 ± 4.85 ^h^	65.80 ± 5.02 ^i^	61.83 ± 5.12 ^l^	53.47 ± 4.23 ^m^	**
Linoleic acid	7.10 ± 0.87 ^a^	7.05 ± 0.45 ^b^	6.85 ± 0.89 ^c^	6.55 ± 0.64 ^d^	5.73 ± 0.36 ^e^	4.21 ± 0.36 ^f^	4.13 ± 0.12 ^f^	4.03 ± 0.88 ^g^	3.78 ± 0.42 ^h^	3.65 ± 0.21 ^i^	2.42 ± 0.12 ^l^	1.23 ± 0.03 ^m^	0.71 ± 0.02 ^n^	**
Arachidic acid	0.26 ± 0.02 ^g^	0.28 ± 0.02 ^g^	0.43 ± 0.03 ^e^	0.52 ± 0.02 ^d^	0.61 ± 0.02 ^c^	0.69 ± 0.32 ^b^	0.78 ± 0.23 ^a^	0.75 ± 0.36 ^a^	0.67 ± 0.04 ^b^	0.54 ± 0.02 ^d^	0.51 ± 0.03 ^d^	0.34 ± 0.04 ^f^	0.16 ± 0.01 ^h^	**
α-Linolenic acid	0.83 ± 0.03 ^a^	0.78 ± 0.03 ^b^	0.75 ± 0.04 ^c^	0.56 ± 0.03 ^d^	0.51 ± 0.01 ^e^	0.47 ± 0.02 ^f^	0.42 ± 0.01 ^g^	0.36 ± 0.02 ^h^	0.30 ± 0.01 ^i^	tr	tr	tr	tr	**
Gadoleic acid	0.13 ± 0.01 ^h^	0.35 ± 0.02 ^c^	0.31 ± 0.01 ^d e^	0.38 ± 0.01 ^b^	0.45 ± 0.01 ^a^	0.33 ± 0.01 ^d^	0.31 ± 0.01 ^d^	0.29 ± 0.01 ^e f^	0.27 ± 0.02 ^fg^	0.26 ± 0.01 ^g^	0.44 ± 0.02 ^a^	0.12 ± 0.01 ^h^	tr	**
OA/LA	4.92 ± 0.56 ^f^	5.25 ± 0.47 ^d^	5.64 ± 0.56 ^c^	6.44 ± 0.47 ^b^	7.37 ± 0.36 ^a^	5.13 ± 0.25 ^e^	4.08 ± 0.86 ^g^	3.88 ± 0.65 ^h^	3.57 ± 0.87 ^i^	3.20 ± 0.12 ^l^	3.16 ± 0.05 ^l^	1.99 ± 0.78 ^m^	0.33 ± 0.01 ^n^	**
∑ SFA	84.43 ± 4.56 ^a^	79.67 ± 5.36 ^b^	78.14 ± 5.25 ^c^	75.64 ± 6.23 ^d^	74.37 ± 5.36 ^e^	73.54 ± 4.21 ^f^	73.4 ± 6.03 ^g^	72.54 ± 4.58 ^h^	71.78 ± 5.23 ^i^	70.26 ± 5.23 ^l^	68.39 ± 4.28 ^n^	64.07 ± 6.12 ^n^	54.49 ± 5.36 ^o^	**
∑ MUFA	7.93 ± 0.14 ^b^	7.83 ± 0.87 ^b^	7.60 ± 0.31 ^b^	7.11 ± 0.69 ^b^	6.24 ± 1.23 ^b^	4.68 ± 0.36 ^b^	4.55 ± 0.74 ^b^	4.39 ± 0.84 ^b^	4.08 ± 0.85 ^a^	3.65 ± 0.45 ^b^	2.42 ± 0.14 ^b^	1.23 ± 0.75 ^b^	0.71 ± 0.54 ^b^	ns
∑ PUFA	10.65 ± 0.23 ^l^	10.17 ± 0.87 ^n^	10.28 ± 0.47 ^m^	10.64 ± 0.84 ^l^	11.92 ± 1.20 ^i^	15.71 ± 1.52 ^h^	16.13 ± 1.36 ^g^	16.52 ± 1.54 ^f^	17.59 ± 1.36 ^e^	19.25 ± 2.03 ^d^	28.26 ± 3.14 ^c^	52.09 ± 3.56 ^b^	76.75 ± 4.58 ^a^	**
MUFA/PUFA	11.26 ± 0.21 ^l^	10.66 ± 0.47 ^n^	10.82 ± 0.56 ^o^	11.10 ± 0.63 ^m^	12.46 ± 0.96 ^i^	16.80 ± 1.09 ^h^	17.11 ± 1.02 ^g^	17.34 ± 1.56 ^f^	18.33 ± 1.22 ^e^	18.57 ± 2.44 ^d^	27.19 ± 2.89 ^c^	50.27 ± 4.58 ^b^	75.31 ± 6.45 ^a^	**
**EVOO_DTD**														
Myristic acid	0.02 ± 00.01 ^a^	0.01 ± 0.00 ^b^	tr	tr	tr	tr	tr	tr	tr	tr	tr	tr	tr	*
Palmitic acid	0.83 ± 0.02 ^a^	0.81 ± 0.02 ^a^	0.78 ± 0.02 ^b^	0.71 ± 0.03 ^c^	0.66 ± 0.03 ^d^	0.53 ± 0.02 ^e^	0.47 ± 0.02 ^f^	0.39 ± 0.01 ^g^	0.37 ± 0.02 ^g^	0.26 ± 0.00 ^h^	0.18 ± 0.01 ^i^	0.08 ± 0.03 ^l^	tr	**
Palmitoleic acid	4.38 ± 0.21 ^a^	4.31 ± 0.22 ^b^	3.76 ± 0.21 ^c^	3.21 ± 0.88 ^d^	2.87 ± 0.45 ^e^	2.86 ± 0.88 ^e^	2.72 ± 0.44 ^g^	2.57 ± 0.55 ^h^	2.56 ± 0.44 ^i^	2.53 ± 0.45 ^l^	2.37 ± 0.45 ^m^	2.26 ± 0.58 ^n^	1.24 ± 0.32 ^o^	**
Margaric acid	0.03 ± 0.01 ^a^	0.02 ± 0.01 ^b^	tr	tr	tr	tr	tr	tr	tr	tr	tr	tr	tr	*
Margaroleic acid	0.76 ± 0.01 ^a^	0.74 ± 0.03 ^a b^	0.72 ± 0.05 ^c^	0.67 ± 0.02 ^d^	0.61 ± 0.02 ^e^	0.55 ± 0.03 ^f^	0.47 ± 0.02 ^g^	0.46 ± 0.02 ^h^	0.42 ± 0.01 ^i^	0.39 ± 0.00 ^l^	0.34 ± 0.02 ^m^	0.28 ± 0.01 ^n^	0.14 ± 0.01 ^o^	**
Stearic acid	3.65 ± 0.25 ^e^	3.94 ± 0.45 ^d^	5.03 ± 0.58 ^b^	5.86 ± 5.03 ^a^	5.87 ± 0.74 ^a^	5.86 ± 0.77 ^a^	5.18 ± 0.88 ^b^	4.88 ± 0.88 ^c^	4.72 ± 0.75 ^c^	4.58 ± 0.78 ^c^	3.87 ± 0.85 ^d^	2.46 ± 0.88 ^f^	1.87 ± 0.23 ^g^	**
Oleic acid	79.92 ± 4.26 ^a^	78.18 ± 5.26 ^b^	75.46 ± 5.23 ^c^	72.75 ± 5.23 ^d^	71.74 ± 4.75 ^e^	71.02 ± 4.22 ^f^	70.77 ± 5.11 ^g^	70.52 ± 4.75 ^h^	69.57 ± 4.23 ^i^	68.4 ± 5.36 ^l^	66.87 ± 5.02 ^m^	64.25 ± 6.23 ^n^	61.67 ± 5.21 ^o^	**
Linoleic acid	7.10 ± 0.23 ^a^	7.09 ± 0.66 ^a^	7.05 ± 0.85 ^a^	6.75 ± 1.22 ^b^	6.69 ± 0.82 ^b^	6.37 ± 0.88 ^c^	5.26 ± 0.45 ^d^	5.09 ± 0.88 ^e^	4.97 ± 0.71 ^f^	4.24 ± 0.88 ^g^	3.48 ± 0.78 ^h^	2.61 ± 0.77 ^i^	1.86 ± 0.22 ^l^	**
Arachidic acid	0.26 ± 0.02 ^i^	0.31 ± 0.02 ^h^	0.52 ± 0.04 ^e^	0.69 ± 0.04 ^c^	0.81 ± 0.03 ^b^	0.81 ± 0.02 ^b^	0.93 ± 0.05 ^a^	0.81 ± 0.05 ^b^	0.79 ± 0.02 ^b^	0.62 ± 0.02 ^d^	0.41 ± 0.02 ^f^	0.35 ± 0.02 ^g^	0.17 ± 0.01 ^l^	**
α-Linolenic acid	0.83 ± 0.05 ^a^	0.81 ± 0.04 ^a^	0.76 ± 0.02 ^b^	0.71 ± 0.02 ^c^	0.67 ± 0.02 ^d^	0.58 ± 0.01 ^e^	0.51 ± 0.01 ^f^	0.47 ± 0.04 ^g^	0.36 ± 0.01 ^h^	0.21 ± 0.01 ^i^	0.10 ± 0.01 ^l^	tr	tr	**
Gadoleic acid	0.13 ± 0.01 ^i^	0.54 ± 0.02 ^e^	0.58 ± 0.02 ^d^	0.67 ± 0.03 ^b^	0.73 ± 0.01 ^a^	0.67 ± 0.01 ^b^	0.61 ± 0.02 ^c^	0.59 ± 0.03 ^c d^	0.47 ± 0.01 ^f^	0.33 ± 0.01 ^g^	0.27 ± 0.01 ^h^	0.14 ± 0.01 ^i^	tr	**
OA/LA	11.26 ± 1.02 ^h^	11.03 ± 0.87 ^l^	10.70 ± 1.21 ^n^	10.78 ± 1.11 ^m^	10.72 ± 0.96 ^mn^	11.15 ± 1.81 ^i^	13.45 ± 1.22 ^g^	13.85 ± 0.93 ^f^	14.00 ± 1.02 ^e^	16.13 ± 1.06 ^d^	19.22 ± 1.22 ^c^	24.62 ± 2.36 ^b^	33.16 ± 4.23 ^a^	**
∑ SFA	4.79 ± 0.12 ^l^	5.09 ± 0.56 ^i^	6.33 ± 0.56 ^e^	7.26 ± 0.99 ^b^	7.34 ± 0.81 ^a^	7.20 ± 0.01 ^c^	6.58 ± 0.85 ^d^	6.08 ± 0.77 ^f^	5.88 ± 0.22 ^g^	5.46 ± 0.84 ^h^	4.46 ± 0.65 ^m^	2.89 ± 0.88 ^n^	2.04 ± 0.21 ^o^	**
∑ MUFA	84.43 ± 5.36 ^a^	83.03 ± 6.23 ^a^	79.8 ± 4.22 ^a^	76.63 ± 4.55 ^a^	75.34 ± 5.88 ^a^	74.55 ± 6.33 ^a^	74.10 ± 6.23 ^a^	73.68 ± 7.23 ^a^	72.6 ± 5.36 ^a^	71.26 ± 4.29 ^a^	69.51 ± 6.34 ^a^	66.65 ± 4.85 ^a^	62.91 ± 3.56 ^a^	ns
∑ PUFA	7.93 ± 1.07 ^a^	7.90 ± 0.87 ^b^	7.81 ± 0.86 ^c^	7.46 ± 1.24 ^d^	7.36 ± 0.73 ^e^	6.95 ± 0.47 ^f^	5.77 ± 0.88 ^g^	5.56 ± 0.88 ^h^	5.33 ± 0.45 ^i^	4.45 ± 0.73 ^l^	3.58 ± 0.74 ^m^	2.61 ± 0.36 ^n^	1.86 ± 0.88 ^o^	**
MUFA/PUFA	10.65 ± 0.98 ^i^	10.51 ± 0.88 ^l^	10.22 ± 1.06 ^n^	10.27 ± 0.82 ^m^	10.24 ± 1.05 ^mn^	10.73 ± 0.61 ^h^	12.84 ± 0.78 ^g^	13.25 ± 0.77 ^f^	13.62 ± 0.81 ^e^	16.01 ± 1.04 ^d^	19.42 ± 1.37 ^c^	25.54 ± 3.02 ^b^	33.82 ± 2.49 ^a^	**

Data are reported as mean ± standard deviation (*n* = 3). tr: trace. Differences were evaluated by one-way analysis of variance (ANOVA) test completed with a multicomparison Tukey’s test. Means in the same row with different small letters differ significantly (*p* < 0.05), ns: not significance; * *p* < 0.05, ** *p* < 0.01.

**Table 9 foods-10-01706-t009:** Antioxidant activity of EVOO, EVOO_DTL, and EVOO_DTDs samples (IC_50_ μg/mL).

Months	EVOO				EVOO_DTL				EVOO_DTD			
	DPPH	ABTS	β-carotene Bleaching Test		DPPH	ABTS	β-carotene Bleaching Test		DPPH	ABTS	β-carotene Bleaching Test	
			t 30 min	t 60 min			t 30 min	t 60 min			t 30 min	t 60 min
0	90.17 ± 4.82	85.43 ± 4.64	105.81 ± 5.61	110.25 ± 5.85	90.17 ± 4.81	85.43 ± 4.62	105.81 ± 5.63	110.25 ± 5.89	90.17 ± 4.82	85.43 ± 4.64	105.81 ± 5.60	110.25 ± 5.18
1	89.45 ± 4.85	83.67 ± 4.55	103.85 ± 5.52	108.12 ± 5.71	81.23 ± 4.44	75.53 ± 4.13	91.64 ± 4.90	95.31 ± 5.18	66.79 ± 3.62	55.38 ± 3.15	71.22 ± 3.92	85.36 ± 4.63
2	89.93 ± 4.81	84.38 ± 4.50	104.02 ± 5.52	108.86 ± 5.72	80.64 ± 4.35	74.71 ± 4.01	91.13 ± 4.98	94.98 ± 5.17	66.46 ± 3.64	54.81 ± 3.15	70.67 ± 3.80	84.71 ± 4.53
3	90.13 ± 4.81	84.79 ± 4.66	104.91 ± 5.53	109.34 ± 5.82	79.72 ± 4.34	71.66 ± 3.91	90.79 ± 4.80	94.13 ± 5.05	65.12 ± 3.69	54.01 ± 3.01	69.73 ± 3.83	84.12 ± 4.57
4	91.67 ± 4.93	85.44 ± 4.61	105.53 ± 5.60	109.97 ± 5.88	78.58 ± 4.24	70.23 ± 3.84	90.36 ± 4.82	93.77 ± 5.08	65.89 ± 3.64	51.23 ± 2.95	69.45 ± 3.83	83.83 ± 4.52
5	92.21 ± 4.91	86.83 ± 4.63	105.79 ± 5.64	110.23 ± 5.84	77.21 ± 4.27	70.68 ± 3.82	88.43 ± 4.71	93.16 ± 5.09	61.12 ± 3.42	48.64 ± 2.73	68.34 ± 3.82	82.49 ± 4.40
6	93.67 ± 5.05	86.12 ± 4.62	106.28 ± 5.66	111.58 ± 5.90	72.64 ± 3.91	65.14 ± 3.64	81.89 ± 4.46	92.44 ± 4.90	60.37 ± 3.32	48.02 ± 2.71	67.88 ± 3.70	82.12 ± 4.42
7	93.92 ± 5.01	87.82 ± 4.75	107.47 ± 5.71	112.23 ± 5.90	71.79 ± 3.93	64.86 ± 3.58	81.24 ± 4.47	91.92 ± 4.90	59.31 ± 3.32	47.58 ± 2.71	65.32 ± 3.62	81.33 ± 4.31
8	95.59 ± 5.12	88.19 ± 4.71	109.26 ± 5.80	114.36 ± 6.01	70.13 ± 3.82	64.21 ± 3.57	80.32 ± 4.39	90.38 ± 4.80	56.69 ± 3.12	46.91 ± 2.74	65.01 ± 3.65	80.25 ± 4.32
9	96.37 ± 5.15	91.14 ± 4.90	110.67 ± 5.81	115.61 ± 6.12	70.87 ± 3.83	63.97 ± 3.52	79.97 ± 4.30	90.12 ± 4.81	54.74 ± 3.04	46.72 ± 2.65	64.88 ± 3.54	79.41 ± 4.36
10	98.36 ± 5.21	92.37 ± 4.90	111.14 ± 5.92	117.44 ± 6.28	68.26 ± 3.75	56.43 ± 3.18	75.83 ± 4.19	80.46 ± 4.32	51.22 ± 2.90	43.83 ± 2.54	61.27 ± 3.48	76.66 ± 4.19
11	99.88 ± 5.32	93.53 ± 5.01	112.31 ± 5.90	118.82 ± 6.25	65.72 ± 3.62	55.79 ± 3.13	75.16 ± 4.15	78.31 ± 4.27	48.33 ± 2.70	39.69 ± 2.34	58.19 ± 3.28	73.58 ± 4.07
12	110.69 ± 5.80	98.46 ± 5.27	112.89 ± 5.90	119.83 ± 6.12	60.47 ± 3.35	51.63 ± 2.94	73.69 ± 4.09	75.42 ± 4.15	46.81 ± 2.62	38.51 ± 2.22	57.86 ± 3.24	71.91 ± 3.95
Sign.	****	****	****	****	****	****	****	****	****	****	****	****

Data are expressed as the mean ± standard deviation (SD) (*n* = 3). Differences within and between groups were evaluated by one-way ANOVA followed by a multicomparison Dunnett’s test (α = 0.05): **** *p* < 0.0001 compared with the positive controls (Ascorbic acid for DPPH and ABTS test with IC_50_ values of 5.0 ± 0.8 and 1.70 ± 0.06, respectively, and propyl gallate for β-carotene bleaching test with IC_50_ values of 0.09 ± 0.00 and 0.09 ± 0.01 at 30 min and 60 min incubation, respectively.

**Table 10 foods-10-01706-t010:** Antioxidant activity of dried “San Marzano” tomatoes samples (IC_50_ μg/mL).

Months	DTL				DTD			
	DPPH	ABTS	β-carotene Bleaching Test		DPPH	ABTS	β-carotene Bleaching Test	
			t 30 min	t 60 min			t 30 min	t 60 min
0	35.23 ± 2.13	23.84 ± 1.52	55.68 ± 3.12	58.37 ± 3.22	35.23 ± 2.16	23.84 ± 1.59	55.68 ± 3.18	58.37 ± 3.22
1	45.23 ± 2.62	31.05 ± 1.93	59.36 ± 3.35	75.23 ± 4.13	40.17 ± 2.32	27.53 ± 1.72	56.26 ± 3.14	69.46 ± 3.81
2	45.96 ± 2.64	31.63 ± 1.95	60.32 ± 3.34	75.89 ± 4.15	40.83 ± 2.34	27.66 ± 1.71	57.34 ± 3.24	70.96 ± 3.83
3	46.58 ± 2.65	32.75 ± 1.98	61.41 ± 3.45	76.73 ± 4.23	41.26 ± 2.44	28.52 ± 1.73	57.71 ± 3.23	71.02 ± 3.90
4	47.61 ± 2.74	33.27 ± 2.02	62.13 ± 3.46	78.22 ± 4.22	41.91 ± 2.41	29.13 ± 1.83	58.26 ± 3.25	71.36 ± 3.91
5	48.64 ± 2.78	34.61 ± 2.02	62.89 ± 3.56	79.34 ± 4.36	42.59 ± 2.42	30.47 ± 1.88	58.40 ± 3.27	73.78 ± 4.05
6	48.97 ± 2.77	35.13 ± 2.15	63.62 ± 3.54	80.76 ± 4.37	43.47 ± 2.58	30.93 ± 1.80	59.12 ± 3.36	74.23 ± 4.08
7	49.48 ± 2.83	36.45 ± 2.16	64.81 ± 3.53	82.12 ± 4.48	45.25 ± 2.60	31.71 ± 1.94	61.89 ± 3.45	76.40 ± 4.16
8	51.31 ± 2.91	38.72 ± 2.25	66.47 ± 3.64	83.43 ± 4.57	45.89 ± 2.62	32.36 ± 1.93	63.23 ± 3.55	77.12 ± 4.24
9	55.73 ± 3.13	41.66 ± 2.47	67.18 ± 3.73	84.17 ± 4.56	46.13 ± 2.61	32.94 ± 2.02	64.36 ± 3.55	78.23 ± 4.22
10	56.69 ± 3.24	42.94 ± 2.58	69.23 ± 3.84	86.64 ± 4.65	46.77 ± 2.72	33.28 ± 2.01	65.73 ± 3.62	78.86 ± 4.25
11	57.12 ± 3.26	44.52 ± 2.53	70.46 ± 3.83	87.29 ± 4.72	47.36 ± 2.73	35.76 ± 2.18	67.52 ± 3.73	79.64 ± 4.35
12	58.44 ± 3.27	46.23 ± 2.61	72.57 ± 3.99	88.51 ± 4.71	48.24 ± 2.76	37.03 ± 2.29	71.48 ± 3.91	82.13 ± 4.41
Sign.	****	****	****	****	****	****	****	****

Data are expressed as the mean ± standard deviation (SD) (*n* = 3). Differences within and between groups were evaluated by one-way ANOVA followed by a multicomparison Dunnett’s test (α = 0.05): **** *p* < 0.0001 compared with the positive controls (Ascorbic acid for DPPH and ABTS test with IC_50_ values of 5.0 ± 0.8 and 1.70 ± 0.06, respectively, and propyl gallate for β-carotene bleaching test with IC_50_ values of 0.09 ± 0.00 and 0.09 ± 0.01 at 30 min and 60 min incubation, respectively.

## Data Availability

All data and materials are available on request to the corresponding author.

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
