# Peer review of "Shelf-Life Evaluation of “San Marzano” Dried Tomato Slices Preserved in Extra Virgin Olive Oil"

_foods, 2021, doi:10.3390/foods10081706_

Round 1

Reviewer 1 Report

The article entitled “Shelf-life Evaluation of San Marzano Dried Tomato Slices Preserved in Extra Virgin Olive Oil”  provides valuable information on the storage stability of an Italian preserve.

I have the following comments for this article:

The article is well written, and the introduction is adequate, the material and methods are satisfactorily discussed, the obtained results are statistically analyzed and presented well. Results are well discussed, and the obtained results support conclusions.

Please correct the typological errors. For instance, in most places, the “β” symbol is missing for the β-carotene. Line 511: complete the parenthesis: “(2.71 mg trolox/g DM for fresh tomato slice vs 3.55 mg trolox/g DM for fresh tomato slice”; “)” is missing

Line 317-333: carotenoids values are discussed in the different units of mg/100g and g/g. For an easy comparison, please change the units to mg/100g.

All the grayscale figures can be replaced with colored figures.

Author Response

Reviewer 1

I have the following comments for this article:

The article is well written, and the introduction is adequate, the material and methods are satisfactorily discussed, the obtained results are statistically analysed and presented well. Results are well discussed, and the obtained results support conclusions.

Q1: Please correct the typological errors. For instance, in most places, the “β” symbol is missing for the β-carotene.

A1: We have checked and corrected.

Q2: Line 511: complete the parenthesis: “(2.71 mg trolox/g DM for fresh tomato slice vs 3.55 mg trolox/g DM for fresh tomato slice”; “)” is missing.

A2: We have checked and corrected.

Q3: Line 317-333: carotenoids values are discussed in the different units of mg/100g and g/g. For an easy comparison, please change the units to mg/100g.

A3: We have done requested corrections.

Q4: All the grayscale figures can be replaced with colored figures.

A4: In order to make the data more readable, we have replaced the figures with tables where it is possible. Where a figure has been inserted, it has been drawn in grey-scale.

Reviewer 2 Report

General Comments

This manuscript discusses the changes in San Marzano dried tomato clices when preserved in extra virgin olive oil (EVOO). In particular, the authors look at specific quality and chemical parameters to compare the tomatoes and the oil properties when stored over time in the dark or in light conditions. Overall, this manuscript is interesting, however, I do have some major concerns relating to their lycopene analysis. There are a few minor typos and errors throughout the manuscript (e.g., missing “beta” and asterisk symbols) that make this difficult to interpret. Although major revision is needed prior to making this manuscript suitable for publication, I am fairly confident that the authors should be able to make the necessary changes.

Abstract

Minor comments:

Line 20: change “carotenoids” to “carotenoid

Line 20:  the symbol for what I assume to be beta is missing

Line 21: change “fatty acids” to “fatty acid”

Line 25-26, this sentence is too long and has too many things going on. Consider breaking the last part into a separate sentence for clarity.

Introduction

Minor comments:

Line 50: Combine this paragraph with the previous line

Line 55: Change “dry” to “dried”

Major comments:

The introduction does a nice job of introducing the context and traditional use of olive oil to store food. However, since this study is looking at evaluating the EVOO quality, CIELab color parameters, phytochemicals (e.g., phenols, cartotenoids), fatty acid profile, and antioxidant activity, it would be helpful if the author introduces some of these in the introduction section. Perhaps provide a more specific background of San Marzano tomatoes specifically, and highlight the specific compounds of interest in them. Also, introducing the potential problem or gap in knowledge just prior to line 62, would strengthen the introduction.

Materials & Methods

Minor comments:

Lines 78-79: clarify that these were San Marzano tomatoes. Were all tomatoes purchased at the same time? Do the authors happen to know more about the tomatoes, farm location, etc?

Line 81: Can the authors specify the thickness of the slices (e.g., in mm).

Line 83-84: Sentence is unclear, can the authors clarify? The EVOO is from an Ottobratica cultivar, but where was the EVOO obtained from? Manufacturer? If it was made by the investigators, a brief description should be included.

Line 91: Rephrase for clarity. Specify what DTL and DTD stand for. Perhaps the authors should rephrase this section to better clarify the conditions of their study.

Line 99: The authors should rephrase “slide tomato” as this does not make sense.

Lines 101-102: Moisture content is a fairly straightforward procedure, but the authors can better clarify here that this was done gravimetrically by measuring the weight loss of the samples after drying for a specific amount of time/to achieve constant weight.

Line 104: Add a “)” after “Italy”.

Line 137: Should the header of this section be changed to be just Extraction, or Phenolic Extraction as it is currently written? Is this extract then subsequently used for phenolic and carotenoid analysis later?

Line 52: Clarify, what is “The sample”? Is this referring to the extract that was described in the previous section?

Lines 165-166: The authors should clarify how the absorbance was converted into carotenoid content (e.g., with Beer’s law, what value was used for molar absorptivity, etc.).

Lines 171-173: This section also describes an extraction process for chlorophyll content, but it seems out of place. Perhaps the authors should move the extraction portion to the previous section and reorganize this existing section for clarity.

Lines 173-174: The authors should clarify the calculation or how the wavelengths were used to calculate chlorophyll content.

Line 175: Missing “beta”

Line 193: The authors could clarify how quantification was done (e.g., linear calibration curves with authentic standards, etc.) and clarify where the standards were purchased from.

Line 221: This paragraph can be combined with the previous sentence.

Line 226: The symbol for alpha is missing.

Major Comments:

For color analysis, did the authors consider also looking at the overall change (e.g., DeltaE)?

I understand that the EVOO is meant to be a control, but did the authors consider having a control for the tomato slices? Does the storage of the tomato slices in the olive oil significantly improve the tomato quality parameters?  

Results

Minor comments:

Line 238: Combine this paragraph with the next

Line 245: Change “help” to “helped”

Line 277: “%” should be added after the moisture content range

Line 280-281: Rephrase for clarity

Line 305- Remove “a” from “significant a reduction of”

Figure 1. Since the figures are shown in black and white, the authors might want to consider re-formatting so that it is easier to distinguish between bars (e.g., choose more contrasting shades)

Figure 1. Caption: “beta” is missing.

Figures 1 – 3: “Sign” is ambiguous. No astersisks are shown on the graph, so as it is currently shown there are no significant differences. The authors should double check all their figures and tables to make sure that the proper significance is shown in the file format.

Table 2: Units should be included

Major comments:

Line 270-271: Do the authors mean “relative humidity” or “moisture content” here?

Table 1: Is one-way ANOVA the best analysis for this? Perhaps Two-way ANOVA would better show the effects of two factors (time and sample type) on the response value.

Line 310-312: Clarify why there was an initial degradation here or why there were differences in lycopene content observed. Lycopene is not a typical compound found in EVOO, so why would there be this present in EVOO without tomatoes? I have seen that other carotenoids (e.g., beta carotene and lutein) have been reported in EVOO. I bring this up because I am a bit concerned about if your HPLC method was able to clearly show/differentiate across carotenoids isolated in this study (C18 columns are known to be a bit problematic for this application and without seeing your chromatograms, I can’t be too sure).

Line 310-312: I also wonder if the authors were able to differentiate between lycopene isomers or if all isomers resolved with the trans peak with this C18 column? C18 columns are known to be a bit problematic for carotenoid analyses, particularly if there are more than one type of carotenoid to be analyzed. It is well known that lycopene isomerizes into cis forms during storage and/or processing. Were the authors able to identify all peaks that appeared in their chromatogram? Since the authors include a lot of supplementary data, perhaps it would be wise for them to also include a representative chromatogram for their HPLC analyses.

Conclusion

Lines 588-590: I’m not sure that the findings directly support this statement. Perhaps the authors need to elaborate a bit more on this to justify.

Author Response

Reviewer 2

General Comments

This manuscript discusses the changes in San Marzano dried tomato clices when preserved in extra virgin olive oil (EVOO). In particular, the authors look at specific quality and chemical parameters to compare the tomatoes and the oil properties when stored over time in the dark or in light conditions. Overall, this manuscript is interesting, however, I do have some major concerns relating to their lycopene analysis.

Q1: There are a few minor typos and errors throughout the manuscript (e.g., missing “beta” and asterisk symbols) that make this difficult to interpret. Although major revision is needed prior to making this manuscript suitable for publication, I am fairly confident that the authors should be able to make the necessary changes.

A1: We have checked and corrected all minor typos and errors throughout the manuscript.

 Abstract

Minor comments:

Q2: Line 20: change “carotenoids” to “carotenoid

A2: We have checked and corrected.

Q3: Line 20:  the symbol for what I assume to be beta is missing

A3: We have checked and corrected.

Q4: Line 21: change “fatty acids” to “fatty acid”

A4: We have checked and corrected.

Q5: Line 25-26, this sentence is too long and has too many things going on. Consider breaking the last part into a separate sentence for clarity.

A5: We have modified sentence as follow: “This enrichment of EVOO by tomato bioactive compounds is reflected in the increase of the antioxidant activity of the oil independently by the presence of light during storage.”

Introduction

Minor comments:

Q6: Line 50: Combine this paragraph with the previous line

A6: We have done requested correction.

Q7: Line 55: Change “dry” to “dried”

A7: We have checked and corrected.

Major comments:

Q8: The introduction does a nice job of introducing the context and traditional use of olive oil to store food. However, since this study is looking at evaluating the EVOO quality, CIELab color parameters, phytochemicals (e.g., phenols, cartotenoids), fatty acid profile, and antioxidant activity, it would be helpful if the author introduces some of these in the introduction section.

A8: As requested the introduction section was improved by adding the concept of food quality, shelf-life parameters etc.

Q9: Perhaps provide a more specific background of San Marzano tomatoes specifically, and highlight the specific compounds of interest in them. In introducing the potential problem or gap in knowledge just prior to line 62, would strengthen the introduction.

A9: We have added information on San Marzano tomato, a protected designation of origin (PDO) variety. This product derived from Agro- Sarnese-Nocerino area (Campania, Italy). We have added into introduction section that this tomato variety has been poorly investigated.

Materials & Methods

Minor comments:

Q10: Lines 78-79: clarify that these were San Marzano tomatoes. Were all tomatoes purchased at the same time? Do the authors happen to know more about the tomatoes, farm location, etc?

A10: Tomato used for this study derived from Agro- Sarnese-Nocerino area (Campania, Italy). We have not indicate the name of the Farm since we bought it in a market in Reggio Calabria (Italy), and did not receive it from the producer. Tomato were purchased at the same time.

Q11: Line 81: Can the authors specify the thickness of the slices (e.g., in mm).

A11: We have added requested information. Tomato slices are 3-4mm thick

Q12: Line 83-84: Sentence is unclear, can the authors clarify? The EVOO is from an Ottobratica cultivar, but where was the EVOO obtained from? Manufacturer? If it was made by the investigators, a brief description should be included.

A12: The study was carried out on the Ottobratica cultivar grown in the experimental orchards belonging to the Calabrian regional government, located near Gioia Tauro in the province of Reggio Calabria, Italy. The olives were harvested by hand in mid-October 2019 directly from the plant, and immediately taken to the laboratory, where the oil was extracted within 12 hours of harvesting, using a crusher “Mini 30” made by Agrimec (Firenze, Italy). After crushing and malaxing of the olive paste, oil extraction was performed at room temperature by means of a pressure system. The olive paste was mixed for 30 min, and then a pressure of 200 atm was applied for 40 min. The oil was separated from the water (by centrifuge), then filtered through a paper at room temperature and analyzed immediately.

Q13: Line 91: Rephrase for clarity. Specify what DTL and DTD stand for. Perhaps the authors should rephrase this section to better clarify the conditions of their study.

A13: We have inserted Table with all acronyms of investigated samples (Please see Table 1).

Q14: Line 99: The authors should rephrase “slide tomato” as this does not make sense.

A14: We have checked and corrected typos.

Q15: Lines 101-102: Moisture content is a fairly straightforward procedure, but the authors can better clarify here that this was done gravimetrically by measuring the weight loss of the samples after drying for a specific amount of time/to achieve constant weight.

A15: The moisture content of tomato slices samples was determined by 934.06 AOAC (AOAC. 2005. Official Methods of Analysis. 18th edn. Association of Official Analytical Chemists. Arlington, VA, USA.). 5 g of tomato samples was weighed precisely into the moisture dishes. The samples were oven dried at 105°C for 3 hr to a constant weight, and then, transferred quickly into a desiccator for cooling. After cooling, the samples were weighed while minimizing exposure to atmospheric conditions. The moisture determina-tion was done in three replicates for each sample. Moisture content was obtained through differences in weight before and after moisture drying to a constant weight. The values obtained were expressed as percentage moisture content.

Q16: Line 104: Add a “)” after “Italy”.

A16: We have checked and corrected

Q17: Line 137: Should the header of this section be changed to be just Extraction, or Phenolic Extraction as it is currently written? Is this extract then subsequently used for phenolic and carotenoid analysis later?

A17: We have corrected title as follow: “EVOO and Dried Tomato Extraction Procedure” since we have used the same extract for all phytochemicals determinations.

Q18: Line 52: Clarify, what is “The sample”? Is this referring to the extract that was described in the previous section?

A18: We have checked and corrected.

Q19: Lines 165-166: The authors should clarify how the absorbance was converted into carotenoid content (e.g., with Beer’s law, what value was used for molar absorptivity, etc.).

A19: The determination of the carotenoids was carried out following the method of Fish et al. 2002 and use calibration curve with b-carotene (Sigma).

Q20: Lines 171-173: This section also describes an extraction process for chlorophyll content, but it seems out of place. Perhaps the authors should move the extraction portion to the previous section and reorganize this existing section for clarity.

A20: We have checked and corrected method for chlorophyll determination.

Q21: Lines 173-174: The authors should clarify the calculation or how the wavelengths were used to calculate chlorophyll content.

A21: We have detailed method applied for chloropyll determination. “The total chlorophyll content was estimated by measuring the absorption at 662 and 644 nm, and calculated according to the procedure previously published by Ašimović et al. with some modifications [23]:

Chlorophyll a = [12.7 (A662) - 2.69 (A644)] x V/100 × w (1)

Chlorophyll b = [22.9 (A644) - 4.86 (A662)] x V/100 × w (2)

Total chlorophyll = [20.2 (A644) + 8.02 (A662)] x V/100 × w (3)

Where, V, Total volume of solution made (mL); w, weight of sample (g); A644, Absorbance at 644 nm; A662, Absorbance at 662 nm. The chlorophyll content was expressed as mg/Kg.

Q22: Line 175: Missing “beta”

A22: We have checked and corrected

Q23: Line 193: The authors could clarify how quantification was done (e.g., linear calibration curves with authentic standards, etc.) and clarify where the standards were purchased from.

A23: The determination of the carotenoids was carried out following the method of Fish et al. 2002 (Fish, W.W.; Perkins-Veazie, P.; Collins, J.K. A Quantitative Assay for Lycopene That Utilizes Reduced Volumes of Organic Solvents. J. Food Comp. Anal. 2002, 15, 309-317. https://doi.org/10.1006/jfca.2002.1069) We have use b-carotene standard from Sigma for beta-carotene quantification whereas for lycopene the  equation reported in attached file was used.

The obtained results in mg/Kg was than converted in mg/100 g.

Q24: Line 221: This paragraph can be combined with the previous sentence.

A24: We have done requested correction.

Q25: Line 226: The symbol for alpha is missing.

A25: We have checked and corrected.

Major Comments:

Q26: For color analysis, did the authors consider also looking at the overall change (e.g., DeltaE)?

I understand that the EVOO is meant to be a control, but did the authors consider having a control for the tomato slices? Does the storage of the tomato slices in the olive oil significantly improve the tomato quality parameters?  

A26: Thanks for comments. We have appreciated suggestion and we have calculated DE*ab values for EVOO_DTL and EVOO_DTD samples during 12 months storage. We have added comments and graph in Figure 1 and Table S3.

Results

Minor comments:

Q27: Line 238: Combine this paragraph with the next

A27: We have done requested correction.

Q28: Line 245: Change “help” to “helped”

A28: We have checked and corrected.

Q29: Line 277: “%” should be added after the moisture content range

A29: We have inserted units.

Q30: Line 280-281: Rephrase for clarity

A30: We have modified sentence.

Q31: Line 305- Remove “a” from “significant a reduction of”

A31: We have checked and corrected.

Q32: Figure 1. Since the figures are shown in black and white, the authors might want to consider re-formatting so that it is easier to distinguish between bars (e.g., choose more contrasting shades)

A32: We have converted the figures into color images to make them easier to see

Q33: Figure 1. Caption: “beta” is missing.

A33: We have checked and corrected.

Q34: Figures 1 – 3: “Sign” is ambiguous. No astersisks are shown on the graph, so as it is currently shown there are no significant differences. The authors should double check all their figures and tables to make sure that the proper significance is shown in the file format.

A34: In order to make the data more readable, we have replaced the figures with tables where it is possible. Where a figure has been inserted, it has been drawn in colour.

Q35: Table 2: Units should be included

A35: We have insert units.

Major comments:

Q36: Line 270-271: Do the authors mean “relative humidity” or “moisture content” here?

A36: We have cheched and corrected.

Q37: Table 1: Is one-way ANOVA the best analysis for this? Perhaps Two-way ANOVA would better show the effects of two factors (time and sample type) on the response value.

A37: We have decided to do One-way ANOVA taking into account that data are statistically analysed also by Principal Component Analysis.

Q38: Line 310-312: Clarify why there was an initial degradation here or why there were differences in lycopene content observed. Lycopene is not a typical compound found in EVOO, so why would there be this present in EVOO without tomatoes? I have seen that other carotenoids (e.g., beta carotene and lutein) have been reported in EVOO. I bring this up because I am a bit concerned about if your HPLC method was able to clearly show/differentiate across carotenoids isolated in this study (C18 columns are known to be a bit problematic for this application and without seeing your chromatograms, I can’t be too sure).

A38: We thanks Reviewer 2 for comments. During the drafting of the work, a method was mistakenly inserted which is not the one applied in our laboratory. The determination of the carotenoids was carried out following the method of Fish et al. 2002, method already applied by our group in other published works such as Teresa Maria Pellicanò, Vincenzo Sicari, Monica Rosa Loizzo, Mariarosaria Leporini, Tiziana Falco, Marco Poiana. Optimizing the supercritical fluid extraction process of bioactive compounds from processed tomato skin by-products. Food Sci. Technol. 2020, 40,  https://doi.org/10.1590/fst.16619.

Once the error was found, the data of the carotenoids reported in the work were re-checked and their correctness confirmed.

Q39: Line 310-312: I also wonder if the authors were able to differentiate between lycopene isomers or if all isomers resolved with the trans peak with this C18 column? C18 columns are known to be a bit problematic for carotenoid analyses, particularly if there are more than one type of carotenoid to be analyzed. It is well known that lycopene isomerizes into cis forms during storage and/or processing. Were the authors able to identify all peaks that appeared in their chromatogram? Since the authors include a lot of supplementary data, perhaps it would be wise for them to also include a representative chromatogram for their HPLC analyses.

A39: The determination of the carotenoids was carried out following the method of Fish et al. 2002 ((Fish, W.W.; Perkins-Veazie, P.; Collins, J.K. A Quantitative Assay for Lycopene That Utilizes Reduced Volumes of Organic Solvents. J. Food Comp. Anal. 2002, 15, 309-317. https://doi.org/10.1006/jfca.2002.1069), method already applied by our group in other case as told you before. Once the error was found, the data of the carotenoids reported in the work were rechecked and their correctness confirmed.

Conclusion

Q40: Lines 588-590: I’m not sure that the findings directly support this statement. Perhaps the authors need to elaborate a bit more on this to justify.

A40: Taking into account Referee suggestion we have decided to remove this sentence.

Round 2

Reviewer 2 Report

I think the authors did a nice job of improving their manuscript and clarifying different aspects.

However, that authors did not address my comment/question about if lycopene "decreasing" and then increasing in EVOO_DTL and EVOO_DTD. I ask this because in lines 348-350, it is unclear why there would be an initial decrease in lycopene content for EVOO that was stored with tomatoes. Perhaps this confusion might be better settled if the authors can clarify (in table and/or section 2.6) if the analysis/extraction for samples EVOO_DTD and EVOO_DTL include extraction on just the oil that was stored with the tomato or the tomato and the oil? If the former, then the authors' justification of lycopene increases overtime make sense, but it needs to be clarified if there is a baseline level of lycopene present in EVOO. If the latter, then the authors would need to better clarify the increases over time (shown in table 3), e.g., if its an improvement of extractability or not.

The authors also should clarify that lycopene values in EVOO appeared to increase over time, although it is not significant (Table 3).

Author Response

Reviewer#2

Q1: I think the authors did a nice job of improving their manuscript and clarifying different aspects. However, that authors did not address my comment/question about if lycopene "decreasing" and then increasing in EVOO_DTL and EVOO_DTD. I ask this because in lines 348-350, it is unclear why there would be an initial decrease in lycopene content for EVOO that was stored with tomatoes. Perhaps this confusion might be better settled if the authors can clarify (in table and/or section 2.6) if the analysis/extraction for samples EVOO_DTD and EVOO_DTL include extraction on just the oil that was stored with the tomato or the tomato and the oil? If the former, then the authors' justification of lycopene increases overtime make sense, but it needs to be clarified if there is a baseline level of lycopene present in EVOO. If the latter, then the authors would need to better clarify the increases over time (shown in table 3), e.g., if its an improvement of extractability or not. The authors also should clarify that lycopene values in EVOO appeared to increase over time, although it is not significant (Table 3).

A1: We appreciate Reviewer ‘s comments and we have modified section 2.6 lines 346-356 as follow ”The lycopene content of EVOO_DTL and EVOO_DTD found at month 0 was probably caused by applying of manual pressure to stack the dried tomatoes inside the jar containing the oil used for preservation. This pressure may facilitate the extraction of lycopene by the EVOO (solvent) as consequence of maceration process, taking into account that analysis was carried out 72 h after jar preparation [32]. Analysis of oils (EVOO_DTL and EVOO_DTD) evidenced that in the first 6 months storage a decrease in lycopene content was observed. Successively, this content increase due the due to the passage of this pigment from the dried pressed tomatoes to the EVOO. After 12 months, an increase in the lycopene content of 0.70 and 0.20% was observed, for EVOO_DTD (0.91-1.59 mg/100 g) and EVOO_DTD (0.91-1.10 mg/100 g), respectively. On the contrary, the lycopene content in DTL and DTD decreased by 0.31 and 0.49%, respectively, with a content variation from 0.98 to 0.67 mg/100 g and from 0.98 to 0.4 mg/100 g, respectively.”

The ability of oil to facilitate the lycopene extraction is confirmed by literature data please see Kehili, M.; Sayadi, S.; Frikha, F.; Zammel, A.; Allouche, N. Optimization of lycopene extraction from tomato peels industri-al by-product using maceration in refined olive oil. Food Bioprod. Proc. 2019, 117, 321-328. https://doi.org/10.1016/j.fbp.2019.08.004.

Moreover, we have checked manuscript for typos and English language.